# Capacity Planning (Capital, Staff and Costs) of Inpatient Maternity Services: Pitfalls for the Unwary

**DOI:** 10.3390/ijerph22010087

**Published:** 2025-01-10

**Authors:** Rodney P. Jones

**Affiliations:** Healthcare Analysis and Forecasting, Wantage OX12 0NE, UK; hcaf_rod@yahoo.co.uk

**Keywords:** capacity planning, parturition, maternity, length of stay, maternity costs, optimum bed occupancy, hospital bed numbers, workforce, Erlang equation, economy of scale, quality of healthcare, health policy

## Abstract

This study investigates the process of planning for future inpatient resources (beds, staff and costs) for maternity (pregnancy and childbirth) services. The process of planning is approached from a patient-centered philosophy; hence, how do we discharge a suitably rested healthy mother who is fully capable of caring for the newborn baby back into the community? This demonstrates some of the difficulties in predicting future births and investigates trends in the average length of stay. While it is relatively easy to document longer-term (past) trends in births and the conditions relating to pregnancy and birth, it is exceedingly difficult to predict the future nature of such trends. The issue of optimum average bed occupancy is addressed via the Erlang B equation which links number of beds, average bed occupancy and turn-away. Turn-away is the proportion of times that there is not an immediately available bed for the next arriving inpatient. Data for maternity units show extreme and unexplained variation in turn-away. Economy of scale implied by queuing theory (and the implied role of population density) explains why many well intended community-based schemes fail to gain traction. The paper also addresses some of the erroneous ideas around the dogma that reducing length of stay ‘saves’ money. Maternity departments are encouraged to understand how their costs are calculated to avoid the trap where it is suggested by others that in reducing the length of stay, they will reduce costs and increase ‘efficiency’. Indeed, up to 60% of calculated maternity ‘costs’ are apportioned from (shared) hospital overheads from supporting departments such as finance, personnel, buildings and grounds, IT, information, etc., along with depreciation charges on the hospital-wide buildings and equipment. These costs, known as ‘the fixed costs dilemma’, are totally beyond the control of the maternity department and will vary by hospital depending on how these costs are apportioned to the maternity unit. Premature discharge, one of the unfortunate outcomes of turn-away, is demonstrated to shift maternity costs into the pediatric and neonatal departments as ‘boomerang babies’, and then require the cost of avoidable inpatient care. Examples are given from the English NHS of how misdirected government policy can create unforeseen problems.

## 1. Introduction

This study is the third in a series investigating international hospital bed numbers, bed occupancy and expressed bed demand [1,2]. This study gives a pragmatic approach to maternity (obstetric and midwife care) planning based on the author’s 30 years of research and experience in wider health care capacity planning. A list of over 200 publications on this topic is available at https://www.mdpi.com/1660-4601/20/24/7171/s1, accessed on 10 January 2024 [1]. Relevant references will be cited using an alphanumeric system as L.2, L.12, etc. in [1].

One of the central issues in capacity planning is the following question: how do we know when we have the optimum number of beds? There is considerable misinformation regarding the optimum average bed occupancy for hospitals [1,2]. One of the key ingredients in capacity planning is the role of queuing theory in determining the number of points of service (beds, midwives, theatres, scanners, etc.) required to deal with the current arriving demand. The Danish mathematician A.K. Erlang developed the Erlang loss function in 1917, and it is widely used and trusted across multiple industries; see L.2–5, L.12 in [1].

Erlang originally used the term ‘server’ as any point where the arriving demand is processed, such as telephone calls arriving at a telephone exchange, call center, or telecommunication satellite, cars at a petrol station, customers arriving at the tills in a shop, tellers in a bank, etc. Both customers and servers can only be a zero or positive number with integer values, although the average arrival rate can be a decimal number. This demand is always expressed as a rate, i.e., per unit of time or per unit of area. The Erlang B equation was formulated to calculate the number of ‘servers’ required to avoid any form of queuing, hence, going elsewhere to obtain service. Erlang B therefore gives profound insight especially in situations where immediate access is required such as in maternity, critical care and several other urgent medical/surgical conditions [1]. Erlang B is most helpful because it links the size of the unit (number of beds), average occupancy and turn-away. Turn-away is the proportion of times that a bed is not immediately available for the next arriving patient, hence the patient either queues as they wait for admission, or their arrival in an ambulance may be diverted to another hospital where immediate access is available [2]. Higher turn-away implies elements of chaos, busyness, patient harm and staff dissatisfaction (see references in [1,2]). Other forms of the Erlang equation are available to handle situations where queuing is allowed, such as a waiting list for elective surgery, and are known as part of wider ‘queuing theory’.

The Erlang equation has been demonstrated to be highly applicable to all aspects of maternity services including fetal, neonatal medicine, pre-birth maternal and delivery, aspects of perinatal care and the networks surrounding large specialist hospitals [2,3,4,5,6,7], and L.2, L.12, L.20–22 in [1].

Erlang B can be used to link the effect of unit size upon average occupancy and turn-away. Turn-away is the proportion of time that a bed or theatre slot is not immediately available. Typically, a turn-away rate of 0.1% or below is required for a functional and safe maternity/midwife unit [2]—see also L.2, L.12, L.20–22 in [1]. Implicit in high turn-away are elements of chaos, inefficiency, staff burn-out, premature discharge and poor safety. For the same average occupancy rate the turn-away rate rapidly escalates as the unit gets smaller. This also explains why smaller units cost more to run; see L.22 in [1].

Queuing theory is largely based on Poisson statistics which describe the natural variation in the frequency of arrivals for integer (whole number) events, i.e., a patient/baby [8]. In Poisson statistics the standard deviation (STDEV) associated with the average arrival rate (arrivals per hour, day, month, etc.) is always the square root of the average. However, at low arrival rates, the distribution becomes increasingly skewed with a minimum of zero arrivals, and the most common arrivals are the average and the average minus one. The lower boundary of zero is compensated for by a tail of low probability but high arrivals. This explains why small units must operate at increasingly lower average occupancy rates to avoid turn-away. Poisson statistics are widely used in the fields of epidemiology and public health.

A recent study using Erlang B has demonstrated huge variation in turn-away at English maternity units [2] with some experiencing alarmingly high levels of turn-away. This raises the question regarding the effectiveness of their planning process, whether any planning process was present.

All pregnant mothers expect that an acute Obstetric unit is available with all the associated diagnostic and surgical facilities—should something go wrong—and this constrains how much care can be shifted into the community without increasing total costs [9]. Indeed, this has profound implications to low population density locations [10].

This study will use examples from various countries and locations to illustrate the steps which a maternity department must take to ensure that it currently has sufficient beds and how many it is likely to need in the future. There is significant emphasis on the factors regulating the local trends for each maternity/midwife unit and the role of uncertainty in future trends. The issue of an optimum length of stay (LOS) will be explored.

The study is primarily aimed at hospital managers and policy makers to highlight the key ingredients for maternity capacity planning, but also highlights issues where academics need to pursue various research questions.

To clarify terminology, in England, maternity units are classified as a Consultant-led Obstetric unit when it is part of an acute hospital site and as Midwife-led for community units. Only the lowest risk births occur at the community Midwife units. In the Obstetric units, most of the care is delivered by midwives, but with Consultant management for the more complex pregnancy and birth-related admissions.

## 2. Materials and Methods

### 2.1. Data Sources

Available English National Health Service (NHS) maternity beds since 1978/79, and quarterly bed occupancy for 2023/24 were obtained from NHS England [11]. Quarterly maternity bed occupancy in Northern Ireland was from [12].

Annual live births and total fertility rate (TFR) in Australia were obtained from the Australian Bureau of Statistics [13]. Monthly and annual birth statistics in England and Wales were obtained from the Office for National Statistics (ONS) [14,15,16,17]. Monthly births in England and Wales were summed into a moving 12-month total. The proportion of English births was calculated each year from [17]. Population projections and the components of change were from the ONS [18].

Live births for English output areas (OA) were obtained from the ONS [19]. Lookup tables to convert each OA to an associated output area code (OAC) were obtained from the Office for National Statistics’ data sets portal [20] and apply to the 2011 census data.

### 2.2. Hospital Episode Statistics

Financial year data regarding all admissions to English NHS hospitals were obtained from NHS Digital (now part of NHS England) via the Hospital Episode Statistics Admitted Patient Care (HES APC) data source [21]. HES APC data include any admission related to pregnancy and birth occurring in Obstetric and Midwife units and the length of stay from admission to discharge which therefore includes any time spent in the birthing unit. On this occasion, the detailed Maternity Services Data Set (MSDS) was not accessed since only high-level trends were required to illustrate various issues. HES APC data was accessed in two ways, namely, at specialty level (Obstetric or Midwife unit) and at primary diagnosis level (3-digit International Classification of Diseases, 10th revision ICD-10) covering ICD-10 chapters O (Pregnancy, childbirth and the puerperium) and chapter P (Certain conditions operating in the perinatal period). Relevant maternity conditions can be coded to ICD chapters other than O and P, but these cover the bulk of the relevant trends. Chapter O (maternity) has 1.1 to 1.4 million admissions per annum depending on the year with the maximum in 2007/08, while Chapter P (neonates) has 150 to 160 thousand per year with the maximum in 2016/17, and Chapter Q (congenital conditions) has 66 to 116 thousand per year with the maximum in 2009/10.

### 2.3. Additional Bespoke Data

Additional anonymized data was kindly provided by two English maternity units. The first was from a large maternity unit and listed 9100 consecutive admissions with date of admission and the calculated overnight stay LOS and the actual real-time LOS. The second was from a medium sized unit which listed the birth weights and admission date of 25,000 consecutive births. The first data set was used in Section 2.4 as part of the process for converting NHS HES APC data into a real-time approximation, and to investigate potential 24-h cycles in admissions and LOS. The second data set was used to illustrate how randomness complicates workload at a daily level even in medium sized units.

### 2.4. Estimating Real-Time Length of Stay (LOS)

In the absence of real-time data for England (only midnight LOS is available) the real number of occupied bed days was estimated as follows:Midnight occupied bed days × 1.035 + sum of same day admissions × 0.5

This calculation allows for 3.5% higher occupied bed days in those patients who stay overnight plus an average stay of 12 h (0.5 days) for all same day stay admissions. These numbers are based on data provided by one of the largest Obstetric units in England during 2017/18 for 9100 consecutive admissions. This unit had 12% of the same day stay admissions with a real-time LOS of 0.37 days (9 h). The England average is 9% and 16% of same day admissions for Obstetric and Maternity units, respectively [21]. The real-time LOS for overnight stay admissions was 4.5% higher than that based on midnight LOS. One hospital cannot be the basis for the English NHS; the numbers chosen for the above formula are a compromise and is only 2% different from the chosen formula. For the same day stay admissions, data are available for all years covering day case admissions; for other same day admissions (elective, emergency, other) the data are available for 2012/13 onward. Before 2012/13, the number in each category was estimated by extrapolating backward from the trend observed from 2012/13 to 2022/23.

The real average length of stay (LOS) was calculated as the estimated real occupied bed days (above) divided by the total admissions. Occupied beds can be calculated as occupied bed days divided by 365 (days per year).

### 2.5. Variability in the Gender Ratio

The variability in the gender ratio (as percent female admissions) for the various ICD-10 diagnoses associated with neonatal admissions (ICD-10 Chapter P), 59 diagnoses, and congenital conditions (ICD-10 Chapter Q), with 82 diagnoses, was assessed to find those diagnoses which show variation which is far higher than those due to chance. The data cover admissions at all English NHS hospitals from 1998/99 onward. Section 2.2 detailed the number of admissions per financial year. Given changes in the number of admissions over time (as per Section 2.2) and the possibility of trends in how events are coded, an index of variability was determined as follows. Both the gender ratio (% female) for each ICD-10 diagnosis (3-digit level) and the number of admissions were tabulated for each year between 1998/99 and 2023/24. The absolute difference in the gender ratio was calculated for successive paired years. One standard deviation (STDEV) of Poisson variation associated with the number of admissions was calculated as a percentage by dividing the square root of the average admissions for the paired years by the average admissions for the paired years. The absolute difference in the percent female gender ratio was then divided by the standard deviation arising from the number of admissions. The median value of this ratio was then calculated. The median value, sometimes called the robust average, was used to avoid potential distortion from years with unusually high values. For example, in P03, the first two years in the time series have unusually high values, and Q99 had three unusually high years. Others such as Q00 show no evidence of unusual values. For some diagnoses such as Q79, an unusually high value in 2020/21 may have been due to COVID-19, although this issue was not investigated further. ICD-10 primary diagnoses were then ranked according to the value of the ratio. This is a dimensionless ratio as both the numerator and denominator are percentages. The minimum value for the ratio in both Chapters P and Q was 0.6, which represents a diagnosis where the variation arises purely due to chance. Only diagnoses greater than 2.5 times this minimum were selected as statistically significant and effectively lie beyond the 95% confidence interval.

## 3. Results

### 3.1. Trends in Available Beds in England

Figure 1 presents the trend in available and occupied maternity beds in England and the ratio of available beds per 1000 births over the period 1987/88 to 2023/24. Available beds in Figure 1 are based on the KH03 statistical return and only cover Consultant-led Obstetric units [11]. Occupied beds include Midwife-led units and are only available from 1998/99 onward [14]. While the trend prior to 2000 can be largely explained by changing attitudes to maternal care leading to declining length of stay, the trend in available beds per birth shows interesting undulations which arise from specific trends in births. Figure A1 in the Appendix B shows the trend in England for the number of occupied beds for Obstetric versus Midwife-led care. Note that because of their smaller size the number of available beds in Midwife-led units will be far higher than the occupied beds. In Figure A1, Midwife-led care only began to expand after 2001/02 through to 2014/15. This tended to displace Consultant-led Obstetric care which reached its minimum value around 2010/11. The difference between available and occupied beds will be explored later.

While this information is interesting it requires analysis to unpack the principles widely applicable outside of England. For example, why was the minimum of 11.4 available beds per birth between 2010/11 and 2012/13 not maintained? Indeed, did it represent a period of bed insufficiency and/or unduly low length of stay (LOS)? Is the current ratio of 13.4 acceptable? The following sections will attempt to detail the multifactorial nature behind right sizing a maternity unit and the issues relating to maternity costs. Preliminary observations [2] relating to maternity bed occupancy will also be expanded upon.

### 3.2. Assessing Current Bed Sufficiency Using Average Bed Occupancy

As in the Introduction, queuing theory and the Erlang B equation provide insight into the issue of bed occupancy in maternity units. If a bed is not immediately available, the patient(s) must queue for admission, or another patient must be prematurely discharged. The number of patients queuing and the delay to admission can be predicted by other forms of the Erlang equations. Figure 2 shows a snapshot of the size (as available beds), average occupancy, and turn-away in the 2023/24 financial year for English consultant-led Obstetric units. As highlighted earlier [2], approximately 2023 represents a point of minimum births in a long-term cycle originating from the World War II baby boom.

For situations requiring immediate access a turn-away of less than 0.1% is desirable, i.e., a bed is not immediately available for one in one thousand arriving patients, while less than 0.001% turn-away covers all possible demand fluctuations which occur during the space of a year. As seen in Figure 2, over half of English maternity units currently meet this critical requirement. One large unit was operating just below 50% turn-away, implying extensive queuing for care and/or premature discharge. Several units were operating near the 20% turn-away line, etc. Units operating below 0.1% turn-away do not have excess beds but are correctly resourced to handle fluctuations in births and peak seasonal demand (see later).

Also note that Figure 2 extends the x-axis down to a fewer number of beds, which will include the smaller birthing units within the maternity unit, Midwife-led units, and neonatal critical care, all of which will have a low average occupancy and/or high turn-away. In the case of the small Midwife-led units, at peak demand the patient will be advised to go to the nearby larger Obstetric unit. Low occupancy implies higher costs per patient, while high turn-away implies low safety. There is a very good reason that the smallest Obstetric unit in England has 16 beds. The smallest Obstetric unit in England is on the Isle of Wight where population size precludes the operation of a Midwife unit.

Note the non-linear relationships in Figure 1 which show that smaller units must operate at disproportionately lower average occupancy—and indeed, higher staff and capital costs per patient; see L.20–23 in [1].

It should be noted that Figure 2 represents a common tool for comparing international bed occupancy in maternity units. Figure A2 in the Appendix B shows the occupancy for maternity units in Northern Ireland (in the UK the NHS in each country of the Union is run independently) with quarterly occupancy over the 10-year period of 2014 to 2024. The scatter in quarterly bed occupancy arises from three factors, namely, longer-term trends to higher/lower births, seasonality in births, and Poisson-related variation, which will be higher in the smaller units. Note the units which have consistently had high turn-away for the last decade—with no apparent attempts to remedy the situation. Hence the rationale for this study. Lower population density in Northern Ireland compared to England leads to generally smaller units, although 15 beds remain the smallest functional size for an Obstetric unit compared to 16 in England.

Figure 2 is also directly applicable to the occupancy levels in the birthing unit and the associated supporting high dependency and critical care units, number of cubicles/beds in the (short stay) maternity assessment unit, and Midwife-led community units [2], L.2, L.20–22 in [1]. Economy of scale factors dictate that major surgery and aspects of critical care for neonates are usually located in larger regional hospitals, while critical care for the mother occurs in the larger intensive care unit covering the whole acute site.

The discussion will cover the issue of how annual and quarterly occupancies (Figure 2 and Figure A2) can be used in the planning process.

### 3.3. Seasonality in Births and Bed Demand

Seasonality is central to all aspects of hospital capacity planning. The issue of seasonality in births, and thus demand for beds, is shown in Figure 3, which uses a count of monthly births in England and Wales between 2010 and 2022 [14].

Monthly births [14] have been divided by days per month to give a comparable daily birth rate. Recall that seasonality in births implies seasonality in conception and the September peak implies conception in December/January—possibly during the Christmas/New Year period.

The fact that seasonality in births affects the average occupancy rate, and hence turn-away, is confirmed by seasonal variation in the average quarterly midnight occupancy for the English NHS as shown in Figure A3 in the Appendix B [11].

Figure A4 in the Appendix B explores how the peak month for births in England and Wales varies from year to year. This variation will be driven by meteorological, social conditions and infectious outbreaks during the preceding December/January period when conception occurs [22].

Note in Figure A4 how the peak month for births has shifted from March to May during the 1930s and 1940s to around September in recent years. The shift toward September appeared to occur in the late 1980s. The reasons for such a shift remain unknown. Also note the high value in September 2021 which could relate to a shift in the prevailing strain of COVID-19 during the earlier December/January period. Issues relating to the effect of infectious outbreaks upon human fertility will be covered later.

Figure 3, Figure A3 and Figure A4 are merely illustrative, and the analyses used in Figure A3 and Figure A4 should be conducted for each maternity unit to reflect specific local environmental and social factors. However, the main point is that the annual average is not a suitable basis for planning since the seasonal peak in demand can be considerably higher in some years than others. The number of available staff also needs to reflect the magnitude (and variability) of the seasonal profile.

### 3.4. Circadian Profiles

To investigate potential circadian cycles a data extract was kindly provided by a large maternity unit which was based on real-time admission and discharge data in the absence of any patient identifiable features. For patients staying less than 24 h (real-time) there was a slight minimum in admissions between 6 and 7 a.m. and a rising trend above 9 p.m. The LOS for these admissions showed a strong 24-h cycle reaching a minimum of a 7-to-8-h average stay between 10 a.m. and 3 p.m. and a maximum average stay of 15 to 16 h from 5 p.m. to midnight.

For patients staying longer than 24 h (real time) there was a large spike at 7 to 8 a.m. which is due to the arrival of women undergoing ‘elective’ C-section (average real time stay of 3.3 days). After excluding this spike there was a distinct 24-h cycle reaching a minimum between 3 to 4 a.m. and a maximum between 11 a.m. and 2 p.m. The average LOS for this group appeared to slightly decline across the day with around a 5-day stay from midnight to 2 a.m. falling to around a 4-day stay between 6 p.m. and 11 p.m. It is unknown how staffing levels may affect the above results. However, these emphasize the concept that capacity planning cannot be effectively achieved using bland averages.

### 3.5. Forecasting Births

Having determined that the current average bed occupancy and local seasonal profile in demand are important short-term factors we must turn to the more difficult issue of longer-term planning. These issues are country and location specific and will be illustrated using several examples. The first example in Figure 4 shows the trends in births in Australia from 1934 to 2022.

The trend in Figure 4 is somewhat complex because Australia has a long-standing policy of immigration. Immediately after the cessation of World War II Australia accepted an influx of refugees from Europe and orphaned children from the UK [23]. In later years, immigration based on occupation has been encouraged and the ethnic composition has changed over time. The fertility rate also shows time trends [13]. Predicting the number of births beyond 2022 will involve multiple (uncertain) assumptions which will differ by state and location. A World War II baby boom is evident, but this is overwhelmed by the pace of migration [23].

The next example in Figure 5 is from England and Wales and shows the trend in births since 1938 plus three birth forecasts by the Office for National Statistics. Note that the 1998-based forecast covered the entire UK and was scaled down to match the total for England and Wales. The UK experienced a pronounced peak in births following the cessation of WWII which leads to a series of peaks and troughs as each cohort grows to reach an approximate common birthing age. The peak around 2012 was amplified by an influx of immigrants from the European Union as the Accession Eight Eastern European countries (Poland, etc.) became eligible for free entry into the UK [24]. The number of births was also markedly affected by the oral contraceptive pill which became widely available in the 1960s [25], hence the trough in 1977.

Note that the 1998- and 2012-based ONS forecasts for births are wildly inaccurate, and the 2020-based forecast can also be questioned. For example, the black dashed line shows a possible underlying trend for the trough in births. Also note that it was pure chance that the 1998-based forecast managed to be close to the actual number in 2021—having failed in every other year.

Figure A5 in the Appendix B demonstrates how the ONS forecast for births in 2035 varies wildly depending on the year in which the forecast is made. Indeed, the forecasts appear to decline directly in proportion to the decline in births after the 2012 peak shown in Figure 5. It is likely that this decline reflects hidden assumptions in the methodology which seem not to have been challenged. Note that the ONS forecast of future births is highly dependent on the assumed total fertility rate (TFR).

It has been observed that in England the TFR shows unexplained systematic variation. This appears to be a wider international problem which is illustrated in Figure A6 in the Appendix B using data from Australia as well as England and Wales. In Figure A6 the absolute value of the moving difference between years has been calculated as a percentage difference relative to the previous year. This is sometimes called the moving range. As can be seen the long-term trend in the moving range shows evidence of a series of peaks and troughs, i.e., systematic variation. Also note that despite the gross differences in births between Figure 4 and Figure 5 there is reasonable agreement between the two countries in A6 to suggest that such trends are international with commonality between developed countries. The relative agreement between the two countries in Figure A6 strongly suggests that it is differences in immigration that drive the main differences in births in Figure 4 and Figure 5.

Regarding the issue of immigration, Figure 6 shows an upward trend in the proportion of births in England and Wales which are for women who were born outside the UK. The dip in 2021 is a direct consequence of the COVID-19 pandemic.

Table A1 and Figure A7 and Figure A8 in the Appendix B illustrate how the country of birth for mothers born outside of the UK can materially affect the trend in the total fertility rate (TFR) [26]. This has implications for capacity planning at the local level where the trend in the composite TFR will depend on the mix of arrivals from different countries of origin, the TFR trends for each country and the uncertainty in these trends. Figure A8 in the Appendix B amplifies the complexity hidden in the overall trend by splitting this down into age bands. The implication is that the trends are highly location specific. The net effect at local level is illustrated in Figure A9 which shows the proportion of births due to parents born overseas in English and Welsh local government areas.

Figure A10 in the Appendix B shows the percentage change in births between 2012 and 2023 for local government areas in England and Wales, with a range from a 42% reduction through to a 22% increase. Note that London boroughs appear in the two tails of the distribution. However, the key point is that migrants will be unequally distributed implying that each maternity unit must construct the equivalent to Figure 6 to inform how the future trends may progress. Although at local level it is the absolute number of births that matter from a capacity planning perspective.

Another way of investigating the trends in births is to use social groups which are like consumer groups and can reveal differences in health behaviours, including choices around births; see B.10–11 in [1]. Figure 7 illustrates the effect of social groups upon the long-term trends in births in England between 2001 and 2019 using the UK Output Area Classification (OAC) [25].

The determination of the social group is conducted at an output area (OA) which is the smallest area to which census data are aggregated. The OAC is a hierarchic classification with eight Super groups, twenty-six groups and seventy-six subgroups. Each OA is allocated to one of seventy-six social groups (the subgroups) using similar methods to those used to derive consumer groups [27]. Note that due to the high ethnic diversity in London there is a specific version of the OAC called the LOAC. On this occasion the name of the group is not important; instead, gross differences in the trends exist between the different social groups. Group six represents the services and industrial legacy group, group three represents the countryside, and group two represents the larger towns and cities [28]. Reference [28] shows a map of social groups across the UK.

The peak in births around 2012 in Figure 7 is thus a composite from all the social groups across England which show differences in the timing and magnitude of the maximum and minimum births. Figure 7 strongly suggests that local area trends in births may significantly differ from the national position. For example, social group 6b1 shows minimal variation in births over time while group 2d3 is the only group which strongly conforms to the pattern in Figure 5, although it peaks earlier than the national average.

Location specific trends in births are driven by the social groups utilizing the local maternity unit. This area is poorly studied. Indeed, a range of social and environmental factors are known to influence human fertility [22] which will be reflected in social groups.

### 3.6. Trends in Admissions Relating to Pregnancy and Childbirth

In this section ICD-10 3-digit primary diagnoses are used to illustrate various trends. The aim is to analyze the data from different viewpoints to inform decision making. Hospital admissions occur during pregnancy and childbirth; however, the ratio of admissions per birth is a useful metric. Figure 8 shows the trend for those ICD-10 primary diagnoses which had the highest ratio of admissions per live birth from 1998/99 onward.

In Figure 8, each primary diagnosis has its own unique trend. Maternal care for suspected fetal problems (O36) shows the greatest increase over time while false labour (O47) showed a rapid decrease between 2007/08 and 2012/13 and thereafter has reached an asymptote. One interesting development is the steady rise in admissions for fetal stress (O68) since the onset of COVID-19. Maternity units will need to assess such trends in relation to present and future workload.

One complicating factor is that over time maternity assessment units have been established resulting in what may previously have been emergency department or outpatient attendances being converted into inpatient admissions. For example, in 2009/10 my own research showed that the ratio of total admissions per birth in different English hospitals ranged from 1.11 to 3.58, with 1.11 likely being a genuine inpatient baseline. In addition, an update of the ICD-10 coding was implemented in 2012/13. However, this only affects a limited number of diagnoses in Chapter O.

Figure A11 in the Appendix B shows the trend in neonatal admissions (ICD chapter P) per 1000 births in England from 1998/99 to 2022/23. Note how this ratio reaches a minimum in 2003/04 and escalates thereafter. The significance of this will be discussed in Section 4.14 in the discussion.

An alternative to admissions per birth is to look at the trends in occupied bed days per 1000 births, as in Figure 9. Occupied bed days are simply the sum of length of stays for all the admissions. Occupied beds are simply occupied bed days divided by 365 days per year. Occupied bed days are ideally calculated using real-time length of stay. In the example given in Figure 9, only midnight occupied bed days were available and therefore a real-time estimate was constructed assuming that all same-day stay admissions had a 12 h stay. In addition, the midnight length of stay has been increased by 3.5% to estimate likely underestimation of the true real-time length of stay.

Note the reduction in bed days associated with single spontaneous delivery (O80). This is partly due to the reduction in admissions for this diagnosis noted in Figure 8 and a reduction in LOS. Occupied bed days for abnormalities of pelvic organs (O34) have dramatically increased since 2011/12. Figure A12 in the Appendix B shows the trend in total maternity bed days per birth. Note that bed days per birth reached a minimum when births peaked in 2011/12 and reaches a maximum of 3.0 in 2001/02 and 2.9 in 2022/23 when births reach respective minimums. A local minimum was reached during the first year of COVID-19 pandemic. Is there an optimum point?

Maternity departments will need to disaggregate trends in admissions and length of stay to attempt forecasts of future bed demand.

Lastly, Figure 10 shows the trend in ‘real’ LOS for several diagnoses which have the highest LOS in 2022/23. Real LOS has been calculated assuming that the actual LOS is 3.5% higher than that calculated from integer midnight figures and that all same day stay admissions have a length of stay of around 12 h.

As can be seen, only one of the selected diagnoses shows a dramatic reduction in the real LOS, namely, placenta praevia. Others are showing an increase. Appendix A shows the trend in estimated real LOS across all 72 primary diagnoses in ICD-10 Chapter O. It is extremely difficult to predict where the trends will go in future years in the face of factors increasing LOS such as obesity [29,30,31,32], age of the mother [33], metal health conditions [34,35] and other risk factors [35] whose incidence varies by social group/location. While the trend down for obstructed labour due to a mal positioned fetus may be reasonably expected, the fact that the trend cannot continue ad infinitum is the issue of importance. Indeed, has the average LOS dropped too low, and at which point did it do so?

Figure A13 in the Appendix B shows the trends in the estimated real LOS for Obstetrics versus Midwife units. As expected, the LOS is lower in the Midwife units due to selection of low-risk births. Average LOS also reaches a minimum around 2011/12 when births are at a maximum.

Another key point from Figure 8, Figure 9 and Figure 10 is that even for national totals, with low statistical error, the trend shows a degree of scatter (uncertainty). At the level of the local hospital the uncertainty is amplified due to the sampling error from a smaller sample and other local factors.

### 3.7. Effect of the Environment on Neonatal and Congenital Conditions

Regarding the role of the constantly changing environment (weather, air pollution, infectious outbreaks) on the susceptibility of the developing fetus to various conditions Figure 11 shows neonatal (Chapter P) conditions where the proportion of female admissions shows higher year-to-year variation than may be expected due to chance.

The gender ratio has the advantage that it is a dimensionless ratio which is unaffected by changes in total births. The method was detailed in Section 2.4. All diagnoses shown in Figure 11 fall beyond the 95% confidence interval.

As seen, year-to-year variation in the proportion female neonatal admissions for other conditions of the integument (P83) and birth injury to skeleton (P13) show the highest variation after adjusting for Poisson variation due to number of admissions. Figure A14 in the Appendix B shows the trend over time for several diagnoses and reveals some interesting long-term trends. High birthweight is trending down, i.e., more males relative to females, low birthweight and problems with temperature regulation are both trending up, i.e., more females relative to males. Such trends would otherwise go completely unnoticed.

This process was also repeated for congenital conditions (ICD-10 Chapter Q) and the top congenital conditions with high statistical significance (from 87 available diagnoses) showing excess variability in the gender ratio were in decreasing order: cardiac chambers and connections (Q20) > other musculoskeletal (Q79) > upper alimentary tract (Q40) > brain (Q04) > lung (Q33) > trachea, bronchus (Q02) > gallbladder, bile ducts, liver (Q44) > anterior segment eye (Q13), Larynx (Q31) > Other spinal cord (Q06).

Subtle environmental forces above that due to Poisson variation are clearly acting to target specific neonatal and congenital conditions. The above diagnoses require investigation to see if the risk is location specific.

### 3.8. An Impending Maternity Crisis?

Both this and the previous study [2] identified that some locations may experience an increase in births over time. Figure 12 is a preliminary attempt to estimate maximum potential bed demand in English maternity units during the month of September (from Figure 2) during the next peak in births likely to occur around 2035 to 2044, assuming the ONS forecast for England may be underestimated (from Figure 4). This is based on current levels of LOS, available beds and average annual occupancy in 2023/24. The projection is based on +5% for the September peak in births (Figure 2) and 21% higher births around 2035 to 2044 (Figure 4). The latter assumption will not apply equally to local maternity units, although it is an excellent example of a capacity stress test. However, under this scenario the English maternity service could be in crisis with 17% of units operating above 99% occupancy which is equivalent to >50% turn-away, and around half of units operating >3% turn-away, i.e., many units are no longer fully functional.

Omitted from the above scenario are the effects of likely trends in obesity (discussed later) and that of the 24-h cycle. There is absolutely no benefit to be gained by attempting to assume that the minimum case will occur around 2035 to 2044. By then it is far too late to make the necessary investment in bed numbers or staff. Every maternity unit in England needs to go through the suggested planning steps shown in this study, and then NHS England needs to determine if there will be enough trained midwives.

As a pragmatic aid to this process a spreadsheet has been provided in Appendix A which allows the calculation of a likely maximum case scenario for maternity units across England and Wales. This can be adapted for other countries as required.

### 3.9. A Survey of Capacity Preparedness in England

To obtain a basic understanding of the level of capacity preparedness in England a Freedom of Information (FOI) request was sent to the 15 hospitals with the highest turn-away (from Figure 2). All were asked three simple capacity planning questions.

Are they aware that the reported bed occupancy at the obstetric unit is higher than may be expected for their size?Has any National or Professional Society guidance ever been published on how to correctly size a maternity unit?Do they have any planning documents relating to the choice of the current number of maternity beds?

All but two were unaware that their bed occupancy was higher than expected for their size. Several others declined to answer the first question because in their opinion ‘high’ occupancy was subjective, or their answer was deliberately worded to be obfuscatory. For question 2 no one was aware that there was any National or Professional guidance, while for the 3rd question no one had any planning documents supporting the current number of beds. One hospital indicated that the situation regarding current bed numbers had been placed on the hospital risk register while another was reconfiguring the maternity unit to increase bed numbers.

One unit was built in the late 1980s, when births were at a maximum. However, in the decade 2011 to 2021 the population in that location had grown by 12%, which could indicate over 50% population growth since the unit was opened. This could explain the current high occupancy with an associated 5% turn-away rate.

Lastly, a technical request was sent to NHS England regarding Midwife numbers in their recently released NHS Long Term Workforce Plan [36]. Namely, “How was the impact of changes in future birth numbers calculated including the effect of immigration?”

No response has been received. In the absence of such, it is likely that NHS England plans for midwife training are not aligned with likely future trends in births.

The overall impression is that there is no maternity capacity planning awareness or competence to be found in the English NHS.

In England comprehensive guidance is available regarding all aspects of the design of maternity services via Health Building Note 09-02. Maternity Care Facilities published in 2013 [37]. While this covers the design aspects it does not cover the issue of how many rooms, beds, etc., will be required.

## 4. Discussion

### 4.1. General Issues

As mentioned in the Introduction the Erlang equation is directly applicable to maternity units. The first reference to the Erlang equation relating to hospital beds in general appeared in 1954 [38] while the first application to Obstetric wards was in 1959 [39]. The methodology should be widely used in hospitals, but widespread ignorance seems to prevail among hospital managers and government agencies.

Over the past 35 years there has been entirely misplaced emphasis in England on building smaller hospitals. This was based entirely on the perception of politicians that the NHS was grossly inefficient and had too many beds [1,2]. Clearly the Erlang equation contradicted this political mantra and unsurprisingly issues such as the Erlang equation and turn-away seemingly never passed the lips of the Department of Health/NHS England. Were NHS managers and the public deliberately left in ignorance?

Figure 2 and Figure A2 gave alternative views for turn-away based on an annual average versus a time series of quarterly averages. The annual average in Figure 2 is useful for a snapshot of the national picture between units. However, the turn-away at each hospital is a general annual average. The longer time series of quarterly occupancies in Figure A2 is useful to see how turn-away is changing over time. At a local level a time series of monthly or weekly occupancies are a useful management tool as part of the wider capacity planning information process.

Also note that the process of planning for capacity in pediatric departments is likewise highly dependent on the (local) trends in births. A recent study showed numerous English pediatric departments having excessive turn-away [2].

### 4.2. Forecasting Long-Term Trends in Births

The Office for National Statistics (England and Wales) emphasizes that population estimates are ‘forecasts’ and not ‘predictions’, since all forecasts rely upon multiple assumptions regarding the future [40]. This is well documented in demographic literature especially when involving migration of younger people, and for the elderly [40,41,42,43]. However, over many years the NHS in England, and most likely elsewhere, has consistently fallen into the trap of treating such forecasts as predictions and then proceeding to make dubious capacity planning decisions [1,2]. The greatest mistake is to take admissions/births by age band for a single year and then make long-term predictions for hospital bed capacity in the absence of any knowledge of how the admissions/births show year-to-year variation and how the admission/birth rate may be trending over time [1,2].

A recent review covering 50 years of healthcare capacity forecasting concluded that there was little retrospective review of the model outputs, leading to complete uncertainty of their long-term validity [44]. Indeed, most only appear to work in the short term. Hence, forecasting births has represented a major emphasis for this study.

It goes without saying that numerous methods, which can include economic cycles, varying sex ratios at birth, educational and migrant characteristics (as will be reflected in social groups), have been applied in attempts to improve the accuracy of such forecasts [45,46,47,48,49,50]. Others have resorted to adaptive machine learning techniques to anticipate the multiple potential causative factors [51]. Unsurprisingly Figure A6 shows that the year-to-year variation in TFR is not randomly distributed around the average. This implies that predicting next year’s TFR will depend on where the current year lies in the cycle of uncertainty. The key observation is that all methods give a different answer, and that complexity and uncertainty abound.

Examples of the trends in births were given in Figure 3 and Figure 4 for Australia and England and Wales. Figure 4 also contained government statistical agency forecasts for future births made over a long-time frame. These forecasts are shown to be wildly inaccurate. The ONS forecasts for England are perhaps overly simplistic with high/low variants based on migration and just one forecast based on estimated TFR. Given the considerable research interest in the topic, it is no wonder that such forecasts are subject to gross failure (as in Figure 4 and Figure A5 in the Appendix B).

In Figure A7 note that the cyclic trend in the TFR for mothers born inside the UK appears to have reached its minimum in 2020 and may be trending upward once again. The trend for women born outside of the UK is itself a composite derived from shifting patterns of migration from different countries. The alternative trend in Figure A7 (black dashed line) may represent the start of an upward trend. The result is that the TFR for future years is an uncertain composite which is unique to each maternity unit location. The key point is to never plan based on the minimum case scenario.

Several comments on the issues in the UK may be helpful. The 2012 peak in births was amplified by the arrival of large number of younger immigrants from the expanding European Union [52]. Because immigration was a contentious topic in the UK the Westminster government publicly downplayed the likely impact and the actual arrivals were 20-times higher than the upper end of the government estimates. No impact assessment was performed on the likely effect on maternity demand. However, it is important to note that the 2012 peak was predictable from past patterns in births arising out of the WW II baby boom—however the exact magnitude was uncertain.

The second key issue is the unreliability of the government statistical agency forecasts (as per Figure 4). Such unreliable forecasts have been made over many years. Clearly the ONS is aware of the multiplicity of issues and seeks expert opinion [53]. However, my experience has shown that forecasts of deaths over the past 30 years have been likewise grossly inaccurate. Given that births and deaths are important components of population forecasting the validity of the future population age structure can be questioned [40,41,42,43]. This will be especially the case among the ages which influence future maternity demand and are most subject to migration.

The ONS recognizes that immigration seems to play the most decisive role and so provides estimates for births based on high and low immigration estimates [54].

The Department for Education also requires data on births and migration to forecast future school requirements. They mainly rely on the ONS forecasts with some adjustments for refugees, i.e., the Ukraine war, and asylum seekers, etc. [55]. They note that a degree of ambiguity is inevitable.

There are several major reasons why the output from the ONS birth forecasting method may contain a hidden flaw. It is known that births occur in teenagers through those aged over 40. The ONS has precise data on all past female births including the nationality of the mother collected during the birth registration process. Hence, for births in (say) 2023 the age of the mother is known, i.e., mothers aged 30 were all born in 1993, etc. Deaths in that birth cohort are known and so the population of females aged 30 is known for 2023 and the number of non-UK mothers is also known via birth registration. Hence, the birth rate for 30-year-olds in 2023 can be calculated [14,15,16,17]. This is repeated over all ages. Likewise births in the future, say 2035, are largely based on past known female births. With 30-year-old mothers being born in 2005, etc. Deaths up to 2023 are known for this cohort but can be reasonably estimated through to 2035. Estimates of migration have improved over the years. The 2012 peak in births will have only reached 23 by 2035 and will reach the highest birth rate age of 30–34 by 2042 indicating that the next peak may endure longer than in 2012. The net result is that the forecast births should not behave in the manner shown in Figure 4 and Figure A5 (in the Appendix B).

Regarding uncertainty in future birth rates a long time series of births per 1000 females by age band is available for England and Wales covering 1938 to 2023 [14] and is shown in Figure A15 (Appendix B). As seen in Figure A15 the birth rate shows undulations over time with age bands 35–39 and 40+ recently reaching a similar value to that back in 1947. The birth rate in 2023 for many age bands is approaching an approximate asymptote. Since 2004 it has been the age bands 25–29 and 30–34 which are the main contributors to total births and neither of these seems likely to show a dramatic reduction in the next decade. Hence births through to 2035 are mainly based on known birth cohorts but with only a small possible reduction in the future birth rate. A method such as an AutoRegressive Integrated Moving Average (ARIMA) or a three parameter model [46] should be able to give reasonable forecasts based on the entire time series. The conclusion is that the ONS forecasts are behaving contrary to how they should and may be improved using readily available methods.

Based on Figure A5, Figure A6, Figure A7, Figure A8 and Figure A9 in the Appendix B this study concurs with the ONS that future births are likely to be heavily influenced by migration, and especially the country of birth of the mother (Table A1) and the age of the arriving mothers (Figure A8). Given the current international instability arising from armed conflicts and high volumes of displaced people, future births in the UK are likely to be considerably higher than the official estimates shown in Figure 4. The situation experienced in 2012 looks to be repeated and midwife training is once more likely to be inadequate as also bed numbers. Figure A9 and Figure A10 show that the trends are highly location specific.

Maternity departments in the UK are strongly advised to make capacity decisions based on the worst-case scenario where the number of births from around 2024 onward will steadily escalate until around 2037 to 2045. The peak in births experienced in 2012 could even be matched in some locations and fewer births could even apply in others.

Maternity departments should not use TFR but rather use a range of pragmatic methods which will include trends in births which can be split by social groups (Figure 7), or the mother’s ethnic group (Table A1). Note that the social group trends in Figure 7 are only illustrative and that the year for the maximum number of births ranges from 2003 for groups 1a2, 1a3, 6a2, 6a3, 6b3, 6b4, through to 2019 for groups 1a4, 1b3, 1c3. The three largest groups (5a1, 5a2, 5a3) peak in 2010 or 2011. The difference between the maximum and minimum births ranges from 83% in 7d4 down to 8% in 6b1, and 9% in 6b3, 6b4, 1c1. The median is 23% as for 3d3.

Another alternative is to look at the trends by electoral ward—which will reflect a mix of social groups. An actual example of such forecasting is available [56].

The construction of new dwellings is also important and is unequally distributed between locations [57]. The local council should be able to provide historical data and estimates for future years. The key requirement is that any chosen scenario should reflect a realistic view of the maximum possible future demand rather than futile efforts to plan based on the minimum case scenario. Such forecasts can then be used to stress test the current bed number as per Figure 12.

Given the uncertainty discussed above, a spreadsheet has been provided in Appendix A which can give an estimate for the maximum case scenario for different locations in England. This maximum case scenario takes the known births in England up to 2023, assumes that 2023 represents the minimum point in the cycle arising from the WWII baby boom, and then assumes that the cycle from 2002 onwards is repeated in 2024 onwards. This profile is then factored down to the equivalent births at a local level using 2008 as the reference point. Local births are compared to this profile and the difference in 2023 is used to adjust the profile from 2024 onward. This is a very pragmatic approach to estimating the maximum case future trajectory for every maternity unit. The future forecast can be modified as actual local data and for England becomes available. This method can be modified to suit other countries.

Four worked examples are provided and range from a maximum possible of births for 2035 to 2044 around that seen in 2012 for Milton Keynes down to 26% fewer than 2012 in Brighton and Hove. These forecasts can then be used to modify the stress tests shown in Figure 12 which can also be updated with the most recent bed occupancy and turn-away data. Future required bed numbers can be calculated to achieve an acceptable turn-away. Note that the stress test process should include any changes in LOS required to deliver a minimum acceptable LOS for singleton and C-section births, etc. These were illustrated in Appendix A.

### 4.3. Seasonality in Births

Seasonality in human births is well-recognized [58]. Season of birth (more correctly season of conception) has also been shown to influence both birth and lifetime risk for several physical and psychological diseases [59,60,61,62,63,64,65]. A study in Czechoslovakia established that birth seasonality was strongly influenced by social determinants. The more educated showed higher seasonality and birth order was also important [60]. Hence Figure 2 regarding births in England and Wales merely confirms this fact. Seasonality in birth weight and average gestational length is also well-recognized [59,61] and may effect local neonatal care demand.

Queuing theory and the Erlang B equation dictate that the maternity unit should be sized to accommodate the point of highest seasonal demand. Based on the work of Bobak and Gjonca [60] it is likely that social group, as per Figure 5, will strongly influence the extent of seasonality experienced at a local level. Midwife-led units may also experience a slightly different seasonal pattern.

### 4.4. 24-h and Weekday Cycles in Bed Occupancy

There is good evidence to show that disrupted circadian rhythms during pregnancy are associated with preterm birth, higher rates of miscarriage and lower birth weight [66]. Studies show that there are peak times during the 24-h cycle where births occur, but that these differ by parity and Midwife versus Obstetric care [67].

Section 3.4 in the results gave evidence from one large English maternity unit that 24-h cycles in both admissions and LOS did exist but were different for those who stayed less than 24 h and those who stayer longer. It was unclear how staffing levels may have affected these results.

The existence of such 24-h cycles is one reason why midnight occupancy levels should not be used in maternity capacity planning. Real-time data must be used. Examples of a 24-h cycle in bed occupancy are available for a medical assessment unit [68], and a small hospital [69]. It is suggested that similar studies be conducted at the local level for maternity units, perhaps using standard 1-h intervals. The existence of such 24-h cycles is a good reason for those maternity units in Figure 2 functioning at an average (midnight) occupancy below the 0.1% turn-away line.

There is also a day of week cycle in births arising from elective C-sections which mainly occur on working days. In England, this can equate to an 18% reduction in daily births on weekends, a 30% reduction during the Easter holidays, and a 40% reduction during the Christmas/New Year holidays. Authors calculation based on [16]. Annual averages are an unsuitable basis for capacity planning.

### 4.5. Lunar and Solar Cycles

A study in North Carolina between 1997 and 2001 and another in Arizona between 1995 and 200 could discern no evidence for lunar cycles in births or birth complications [70,71]. A study in Germany between 1920 and 1989 rejected a role for lunar cycle but did note a small role for sunspot numbers [72].

A detailed study in Japan did however reveal that the proportion of babies born at night did rely on the lunar cycle [73]. This concurs with the 24-h cycles reported in the previous section and may influence the issue of day/night capacity planning. Hence the suggestion that occupancy should be documented at hourly intervals using real-time rather than midnight data.

### 4.6. Infections and Pregnancy

There are over 3000 known species of human pathogen, see I.3 in [1] and probably far more than 30,000 relevant strains and variants. This includes persistent and transient infections. The majority of these are entirely unresearched in terms of their transient through to permanent clinical effects upon fertility (male/female), and the mother and fetus. While it is usual to think of infection in terms of obvious clinical symptoms the reality is far more nuanced.

Infection of humans by pathogens triggers the processes of pathogen interference which are primarily regulated by interferons, see I.3 in [1], and which in turn are regulated by noncoding RNAs (ncRNAs). ncRNAs regulate gene expression and hence numerous diseases [74]. The host will have a unique profile of ncRNAs influenced by the environment and various conditions while each pathogen also has a unique ncRNA profile [75,76]. The resulting clash of ncRNA profiles can presumably influence fertility and pregnancy outcomes [77,78]. Hence, various infections can transiently influence fertility in both men and women [79,80] and miscarriage [81,82]. The combined effect should be transient dips in the fertility rate. Transient increases in fertility have also been documented, see S.9 in [1].

Pregnancy is a form of precisely timed immune manipulations [83,84] with additional metabolic changes [85]. Many of these changes are regulated by small noncoding RNAs [86]. Hence, the range of adverse fetal outcomes associated with infection during pregnancy [82].

The totality of the workload experienced in the maternity unit will fluctuate with local, regional or national outbreaks of the 3000 known species of human pathogen. There is ample evidence that previously uncharacterized infectious events have operated at local, regional and national level, see D.8, Q.1–18, R.1–17, S.9 in [1] which will have knock-on effects to admissions and workload at local level. Highly nuanced mechanisms such as ncRNA profiles are required to explain some of the volatility in the gender ratio for neonatal and congenital conditions detailed in Section 3.7. Such poorly studied and highly nuanced health behaviours should show social group and location-specific prevalences.

### 4.7. Risk Factors

During pregnancy and birth there are a range of risk factors for health of the fetus and mother which increase costs [29,30,31,32,33,34,35]. Obesity is one of the most recognized risk factors during pregnancy and for adverse pregnancy outcomes [29,30,31,32,35]. Obesity also leads to longer average LOS and higher costs [30,31,32,35]. The level of obesity is increasing in every country [29].

In developed countries the average age at birth is increasing and had reached 30.9 years in England and Wales by 2021 [15]. Older women are at greater risk of birth complications and increased length of stay [32,87]. Each maternity unit will need to assess how local trends are likely to affect future demand in terms of complications during labour and increased LOS.

Social group (as in Figure 5) is indicative of health behaviours and is likely to be associated with levels of obesity, alcohol and drug abuse, smoking, and poor nutrition, which affect the level of complexity during birth [88,89,90,91], and hence especially that experienced at a local level. Most countries will have some form of social group classification, and analysis of length of stay, seasonality and 24-h cycles in occupancy based on social group will summarize the reasons for local deviation from national averages. In the absence of social groups forecasts can also be made using ethnic groups and/or by electoral wards. One example of such an approach is available [56].

While daily and seasonal variation in demand has been noted it is important to realize that the external environment (absolute levels and variations in air pollution, temperature and infectious outbreaks) introduces additional volatility in maternity demand.

The issue of unexpected volatility was explored in Section 3.7 where the year-to-year volatility in the gender ratio for certain neonatal and congenital diagnoses was shown to be far higher than could arise from chance alone. It is widely recognized that certain conditions occur more frequently according to sex, however, environmental causes for variation in the resulting gender ratio are poorly understood.

As an example, it is known that the gender ratio at birth is sensitive to the background level of radiation, including radiographers, see Q.4, S.9 in [1]. In addition, my own unpublished research has shown that the gender ratio at birth for different locations in England is sensitive to the background levels of the radioactive gas radon. Radon levels vary considerably due to ground geology [92], and the levels of radon are also reflected in the incidence in lung cancer [93]. Two studies are available linking higher radon levels to the incidence of low birth weight and hypertensive disorders in pregnancy [94,95].

Hence, the insistence in this study that maternity demand, in all its complexities, be investigated at local level with a suitable low turn-away level to cope with all possible fluctuations in demand.

### 4.8. The English NHS National Maternity Review

It is apposite to take a pragmatic view of one example of a national maternity review. Such a review was published in 2016 covering England which made a host of recommendations based on patient focussed care [96]. However, this review contains no mention of the role of bed occupancy and turn-away in patient safety (as in Figure 2), and by implication matters of size, population density and distance; nor does it give any advice regarding the issue of future trends in births and the capacity planning necessary for bed and staff numbers. Based on the recommendations of the National Maternity Review, the Maternity Transformation Program was implemented across England [97] and a three-year plan for improvement was published in 2023 [98]. However, this plan omits addressing issues around bed capacity planning, the need to maintain safe levels of turn-away, and seemingly, issues of a minimum acceptable LOS for certain types of care.

Another review of the English NHS published in 2024 by the newly elected Labor government also makes no reference to the adverse effects of high turn-away but makes an oblique reference to problems with patient flow [99], which is the direct outcome of turn-away. Patient flow is difficult to quantify while turn-away can be specifically quantified.

### 4.9. Issues of Population Density and Distance

Access to different types of maternity services shows a distinct urban/rural divide largely driven by population density [10,12,100,101,102,103], which will have profound effects upon economy of scale.

A 2011 study in England gave a map of the locations for different types of maternity care [102] and emphasized that location profoundly influences what can be feasibly accessed—mainly due to population density. Spatial access is profoundly important in perinatal care [103]. A one-size-fits-all approach is not possible.

### 4.10. Does Decreasing Length of Stay (LOS) Actually Reduce Costs

In a 1996 review of this contentious topic, it was stated that reducing LOS only has a marginal effect on costs for the following key reason [104]:


*“This is because for both medical and surgical patients, the main costs occur in the first half of the stay when input from staff, investigation, and intervention are at a maximum. Stays in hospital are almost always shortened by reducing lower dependency “cheaper” days, usually in the second half of the stay”. Along similar lines another study noted that “not all hospital days are economically equivalent”*
[105]

Others have noted that there are wider community economic costs associated with shorter LOS which are rarely included in the calculation of total cost [106]. Another study noted that a significant proportion of the supposed cost of an inpatient stay is due to fixed costs, over which the unit has effectively no influence [107]. These costs do not go away and are divided by fewer days of stay to perversely inflate the calculated cost. Table A2 in Appendix B gives a partial list of supporting acute hospital functions whose costs must be apportioned to the maternity and other patient facing departments. How such costs are apportioned will vary between countries and individual hospitals. Countries with high administrative costs such as the USA will have a much higher percentage of apportioned costs and the Figure of up to 60% of overhead costs quoted in the Abstract comes from a US trauma center [105].

Older references have deliberately been cited because the fundamental issues have been well documented over many years. None of these dispute the fact that cost reductions can and should be made; however, the promised reductions in cost never fully materialize and most often staff workload is increased as more patients are crammed into the available beds [107]. This latter point returns us to the issue of hospital bed occupancy and the undesirable effects of turn-away (Figure 2).

Based on the above, maternity departments are urged to work with their finance department to understand the following:How are direct maternity costs allocated based on time? Is time-based costing used?How are wider hospital overhead costs allocated to the maternity department, and would a change in the apportionment method used for each function have a significant effect on the supposed ‘cost’?How do costs behave over time (fixed and variable costs, marginal costs, step increases/reduction due to changes in demand)?

As an example, it may surprise the maternity department to learn that the capital cost, known as depreciation, of a new administration block or surgical unit is generally apportioned ‘equally’ across all patient facing departments, and that the method of apportionment can vary, i.e., square meter of buildings, number of admissions, patient bed days, etc. Each method will give a different level of ‘cost’ allocated to maternity.

### 4.11. Pitfalls in Benchmarking Length of Stay (LOS) and Costs

There is a huge variation in different aspects of postpartum LOS between countries [108], suggesting that different philosophies regarding optimum care are common. For 92 countries it was observed that the average postpartum LOS ranged from 1.3 to 6.6 days, 0.5 to 6.2 days for singleton vaginal deliveries, and 2.5 to 9.3 days for cesarean-section deliveries. They assessed that the percentage of women staying too short a time ranged from 0.2% to 83% for vaginal deliveries, and from 1% to 75% for cesarean-section deliveries [108]. The UK was noted as a high-income country with the shortest average LOS for singleton vaginal birth.

Due to an acute shortage of hospital beds coupled with underfunding relative to demand [1,2], healthcare services in England are obsessed with achieving the minimum possible LOS, and this thinking may have encroached upon maternity LOS trends.

Hospitals in England have a wide range of average LOS for each diagnosis, Healthcare Resource Group (HRG) or Diagnosis Related Group (DRG). HRGs are exclusive to England and were introduced in the early 1990s [109]. In many instances the low and high examples of LOS counterbalance each other and arise from ambiguity in the process of clinical coding. For each patient, the depth of coding, i.e., the average number of codes per admission, can vary wildly between hospitals, complications during surgery can be inadvertently or deliberately omitted, diagnostic ambiguity can cloud the recorded primary diagnosis, etc. In some instances, hospitals have deliberately manipulated the clinical coding process to conceal poor care.

Both HRGs and DRGs rely on the assumption that the local hospital is at the national average for the proportion of diagnosis and procedure codes within each HRG/DRG. This assumption is often invalidated—indeed it is deliberately manipulated via upcoding when attempting to conceal poor care. In the USA, upcoding has become an industry with supporting software to achieve this goal. Between 2011 and 2019, the share of US hospital discharges that were coded as the highest severity increased by 41%, of which 29% were associated with upcoding [110]. Another study noted that failure to fully code for sixteen chronic conditions accounted for 22% of the 2020 Medicare/MA risk-score gap [111]. A similar situation likely exists in maternity departments around the world.

There does not appear to be any benchmarking tool which considers the average occupancy and hence the turn-away of the hospitals in the benchmark reference group. Specifically in the case of Maternity, it is likely that high-turn-away units have artificially low average LOS due to premature discharge. It is also highly likely that the high turn-away units may be shifting neonatal care, which should occur in the maternity unit into emergency department attendances or pediatric admissions [112,113,114].

In addition to the above, the use of LOS calculated at midnight is a relic from the days when the matron would visit the wards around midnight and write on a sheet of paper the number of available and occupied beds. Midnight LOS gives misleading averages especially when there are large numbers of same-day stay admissions, which have a midnight LOS of 0; see K.1–K.9 in [1]. LOS should reflect national standards for good patient care, not a desperate race to the bottom.

The best summary for this section comes from the Abstract in the study by Bowers and Cheyne which was published nearly 10 years ago in 2015 [114]. The full quote is as follows:


*“Reducing the length of time women spend in hospital after birth implies that staff and bed numbers can be reduced. However, the cost savings may be reduced if quality and access to services are maintained. Admission and discharge procedures are relatively fixed and involve high cost, trained staff time. Furthermore, it is important to retain a sufficient bed contingency capacity to ensure a reasonable level of service. If quality of care is maintained, staffing and bed capacity cannot be simply reduced proportionately: reducing average LOS on a typical postnatal ward by six hours or 17% would reduce costs by just 8%. This might still be a significant saving over a high volume service however, earlier discharge results in more women and babies with significant care needs at home. Quality and safety of care would also require corresponding increases in community based postnatal care. Simply reducing staffing in proportion to the LOS increases the workload for each staff member resulting in poorer quality of care and increased staff stress.”*


### 4.12. When Did England Reach the Optimum LOS?

One example of a patient-centered definition of the optimum LOS was published by the American Academy of Pediatrics in 2015 as follows:


*“The hospital stay of the mother and her healthy term newborn infant should be long enough to allow identification of problems and to ensure that the mother is sufficiently recovered and prepared to care for herself and her newborn at home. The LOS should be based on the unique characteristics of each mother-infant dyad, including the health of the mother, the health and stability of the newborn, the ability and confidence of the mother to care for herself and her newborn, the adequacy of support systems at home, and access to appropriate follow-up care in a medical home. Input from the mother and her obstetrical care provider should be considered before a decision to discharge a newborn is made, and all efforts should be made to keep a mother and her newborn together to ensure simultaneous discharge”*
[115]

In the USA, the pressure by insurance companies to reduce postpartum LOS (and costs) led to a situation where around 2000 various states passed legislation to regulate this. In California, three years after the passage of the postpartum hospital stay legislation, the rate of neonatal readmission had reduced by 20%, while that for neonatal infections had reduced by 30% [116].

Figure A11 (neonatal admissions per birth) seems to indicate that England reached the optimum LOS for the minimum neonatal readmissions in around 2003/04. This LOS must be understood in the context of the average age and parity of mothers, along with the trends in obesity, etc., which have occurred since 2003/04. Hence, the optimum LOS is almost certainly drifting upward with time. The optimum LOS will also probably vary by social group (Figure 7), since social group reflects ethnicity and health behaviours. Hence Figure 9, Figure 10 and Figure A13 must be interpreted relative to the years 2003/04.

The final issue in this section is the question as to whether governments should stipulate minimum acceptable LOS for aspects of maternity care. As noted above, such standards were introduced in the USA. As noted above, the comprehensive international study by Campbell et al. [108] concluded that the percentage of women staying too short a time post-delivery in various countries ranged from 0.2% to 83% for vaginal deliveries and from 1% to 75% for cesarean-section deliveries. The median value of LOS for singleton vaginal delivery was around 2.7 days, ranging from 0.5 days in Egypt to 6.2 days in Ukraine [108]. In England, the LOS for a single spontaneous delivery fell from 1.8 days in 2003/04 to 1.2 days in 2022/23. See Appendix A. Single delivery by C-section fell from 4.3 days in 2003/04 to 1.7 days in 2022/23. LOS has declined since 2003/04 in 58 out of 72 current ICD-10 diagnoses. In the above section, 2003/04 was suggested to be the LOS for minimum neonatal admissions.

It is of interest to note that the real-time LOS of 3.3 days for an ‘elective’ C-section in Section 3.4 for a large maternity unit in England strongly suggests that the current LOS of 1.7 days across England has nothing to do with patient benefit, but instead is a dash to minimum LOS to reduce perceived ‘costs’ irrespective of the impact upon mother and baby.

It is strongly suggested that governments investigate which diagnoses should have a minimum standard for LOS.

On the other hand, the spreadsheet in Appendix A shows that LOS has increased for some disorders such as hypertensive disorders with proteinuria (O11), malnutrition (O25), multiple gestational complications (O31), amniotic and membrane disorders (O41), umbilical cord complications (O69) and puerperal sepsis (O85)—which require further investigation with respect to causes, i.e., whether these specific risk factors are highest among particular social groups, etc.

None of the above ignores the duty of maternity units to reduce costs arising from poor care which will occasionally lead to patients with extended LOS. There is no substitute for a learning culture where mistakes can be openly discussed and learned from. This is seemingly difficult to achieve in an environment of chronic underfunding which forces managers to make irrational and counterproductive decisions. Even in the absence of a learning culture it is always beneficial to focus on the tail of very long stay patients to determine likely causes.

### 4.13. Matching Staffing with Demand

In England, the commercial tool Birthrate Plus^®^ (Birthrate plus, 124 Thorpe Road, Norwich, Norfolk, UK) allows Maternity departments to implement the National Institute for Health and Care Excellence (NICE) safe maternity staffing guidelines [117]. Many countries will have similar guidelines and supporting tools.

However, Birthrate Plus^®^ does not attempt to forecast future births and does not address the issue of the required number of beds.

As outlined in this study, Maternity units are recommended to create birth/demand scenarios to determine the range of staffing needed for planning and risk mitigation. In 2018, NHS Improvement (now part of NHS England) detailed a plan to achieve safe staffing levels in the NHS in England [118]. However, there is no accompanying plan to achieve safe levels of hospital bed numbers and associated turn-away. Three decades of flawed bed planning in the wider NHS has made this difficult [1,2]; however, there is room to achieve such a goal for maternity and pediatric services.

### 4.14. Size, Statistical Chaos and Income

In all the national trends shown in this study, despite being based on very large numbers of births and admissions, there is scatter around the trend line. In national data, Poisson-based statistical scatter is minimized leaving the residual systematic or environment-based variation. If we assume just 100 maternity units in the entirety of England and Wales, this gives around 6000 births per annum, 500 per month or 16 per day.

As the size of the maternity unit decreases, the Poisson-based variation begins to dominate. For example, in Figure 3, one standard deviation (STDEV) of Poisson variation at national level is only ±0.45%; however, in the medium sized unit—which submitted daily data—there were 425 births per month, and 1 STDEV of Poisson-based scatter was at ±4.9%, which has overwhelmed the seasonal profile in Figure 3, and has demonstrated that seasonality has almost become irrelevant to capacity planning at the local level.

In the medium sized unit, there was an average of 13.9 births per day ±3.7 (STDEV) with actual daily births ranging from 1 to 27 at the statistical extremes. The case mix for those births will show wild extremes in complexity and hence workload and LOS. For example, the average daily birth weight ranged from 2656 to 3740 g with maximum extreme in birth weight from 220 to 5600 g (authors calculation based on actual data). As it were, statistical chaos reigns. Hence, ‘safe’ staffing becomes a somewhat academic concept and theoretically only ‘works’ using annual averages.

The above-mentioned wide variation in daily births, assuming that the unit is staffed to handle the daily average of 13.9, raises interesting questions regarding how medium sized units cope with very high demand days [119]. One study demonstrated that C-section rates rise as the ratio of staff-to-births declines [120]. Another study showed that high volume days, defined as above the 75th percentile of daily delivery volume, was associated with increased risk of several maternal and neonatal complications [121]. The 75th percentile for the medium sized hospital is 17 versus the average of 13.9 per day. This will disproportionately affect smaller maternity units. It is of interest to note that English maternity units tend to be larger than in other countries [119]. ‘Safe’ staffing in maternity units needs to be determined based on size.

Let us extend the concept of statistical chaos to the income received by such a unit based on an HRG/DRG tariff. One such simulation has been conducted for a much larger whole hospital with around 31,000 admissions per annum, and even for a large hospital the extremes of income are huge—which may not match with the costs incurred; see N.2 in [1].

Over the past 30 years, the deficiencies lying hidden in the English HRG tariff have led to a long list of ‘distortions of reality’; see O.1–21 in [1], In England, government policy implementation always prevails over reality. The government only sees the big numbers and has no idea of the impact and imposed hardship at the local level.

There is no solution to this conundrum but serves to point out that maternity units are very small in the grand scheme of things and are therefore subject to considerable financial and operational risk; see N.1–39 in [1].

### 4.15. Fair Funding for Maternity Units

In England, the HRG tariff was seen as a fundamental support to the Purchaser/Provider split introduced by the Thatcher government [109]. It was declared that each HRG would have a single cost applicable to all hospitals. Clearly this contradicted the universally acknowledged reality of economy of scale. Hence, for many years the Department of Health maintained that HRGs did not show evidence for economy of scale. This was ‘true’ in a deliberately obfuscatory way, only because the costing and pricing process in NHS hospitals was exceedingly poor, see N.1–15 in [1]. However, if costs were aggregated at the specialty level, economy of scale was clear (authors unpublished analysis), and this has been confirmed by others [122,123]. Other research shows that economy of scale is present at the HRG chapter level, which is most prominent for unscheduled care [124].

Despite refinement in the rules and advice regarding how hospitals calculate the prices used in the HRG tariff [125], the reality is that most hospitals have insufficient resources in their costing and pricing teams to ensure genuine like-for-like inputs; see O.19 in [1].

In 2013/14, a maternity pathway tariff was introduced with a per woman system of single payments for each stage of pregnancy and childbirth, namely, antenatal, delivery and postnatal [126]. This replaced the previous fee per episode system where some hospitals were counting outpatient procedures as ‘inpatient’ to increase their income. The antenatal and postnatal segments have three levels of payment (standard, intermediate, intensive), while delivery has two levels depending on complications and comorbidities. This pathway tariff does not recognize the role of size on costs.

Indeed, given the fact that there are over 70 ICD-10 3-digit primary diagnoses with real LOS ranging from 0.7 days (spontaneous abortion, O20) through to 5.7 days (pre-eclampsia on chronic hypertension, O11), it is likely that the pathway tariff may require more categories. This is because the assumed national average case mix in each category may not apply across all locations with their divergent mix of social groups.

Figure 2 shows that maternity units cannot maintain a uniform occupancy level and must therefore experience unavoidable differences in staffing and capital costs. This is not a new observation; see O.21 in [1].

This implies that the only way to provide fair funding for maternity units is to adjust the HRG/DRG tariff of prices based on the units’ size as per the highly nonlinear relationship seen in Figure 2. Such an adjustment is readily calculated; however, it must also be adjusted so that all are compared at equal turn-away. For example, from Figure 2 we see that a 16-bed Obstetric unit has a reported 76% annual average occupancy, which is higher than that for a unit with 109 beds and an 85% average occupancy. The management at this hospital have felt pressured to run the Obstetric unit at high average occupancy to compensate for lower income inherent in the HRG tariff compared to the real costs.

The HRG tariff wrongly assumes that all units operate at the national average. In English Obstetric units, this implies 52 available beds operating with 33 occupied beds, or a 63% average occupancy. This lies between the 0.1% and 0.01% turn-away lines, which is around the recommended annual average occupancy; hence the large cluster of small units operating above 63% average occupancy in Figure 2. They have been forced over many years to sacrifice appropriate turn-away (and likely safety) to claw back their underfunding implied by the HRG tariff. On the other hand, the cluster of larger units operating above 63% average occupancy are making a profit at the expense of appropriate turn-away and potentially lower associated safety. This reality also applies to every specialty in a hospital, especially those which are small, i.e., paediatrics [2].

Such a cost adjustment is already applied to local income via the Market Forces Factor (MFF). The MFF adjusts local costs for the ‘unavoidable’ costs of doing business (land, buildings, business rates, salaries) [127]. The basic HRG tariff is then adjusted via the MFF for that location. An economy of scale factor (ESF) is required.

It should come as no surprise to note that Scotland, Wales and Northern Ireland have all abandoned the purchaser/provider split and the HRG tariff.

The government policy instrument of the HRG tariff has become the direct source of unfair funding and likely the source of unsafe practices. It is not a wise strategy to distort reality to implement policy.

### 4.16. Flexible Staffing to Offset the Efect of Size

As the size of a maternity unit decreases, the fluctuations in the workload become more severe. See L.20–22 in [1]. The only way to counter this is to staff the unit in a highly flexible way. This implies a core number of full-time staff supplemented by a pool of ad hoc staff willing to work at short notice; see L.31 in [1]. This would imply greater cooperation between nearby units and a system of mobile phone alerts to see who is currently available.

As discussed above, the workload is most likely to be higher at night or on workdays. Such a system may be attractive to recently retired midwives or those not wanting to work full time. Unfortunately, such schemes are most readily implemented in large cities where maternity units are already larger than average. However, there are limited other alternatives to minimize staff costs in smaller units.

Another possible solution is to combine the Obstetrics, Gynecology, and female Urology beds into a women’s unit. Gynecology and female Urology would be at one end and Obstetrics (mainly the birth aspects) at the other, with a middle section able to accommodate the capacity surges focusing on the non-birth aspects of Obstetrics. An analysis of past trends in bed occupancy would determine the respective splits. The idea is to increase the size of the total bed pool to allow a higher average bed occupancy across the larger pool.

It is likely that these suggestions are unworkable, however, they illustrate the limited options available to cope with the reality of the unavoidable effects of economy of scale.

### 4.17. A System-Wide View of Maternity Costs

In countries where there is a purchaser/provider split, the purchaser will often fall into the trap of thinking that they can save money by shifting maternity care out of the ‘expensive’ Obstetric unit into purchaser-run community midwife units and home births. They assume that the ‘expensive’ Obstetric unit will still be available when things go wrong in their ‘low cost’ community units, where adverse outcomes have increased due to pressure to shift as much care as possible (arbitrary percentage targets, etc.) out of the Obstetric unit.

The flaw in this thinking is that shifting care out of the Obstetric unit sends that unit down the economy-of-scale curve in Figure 2, and still leaves the acute hospital overhead costs to be apportioned. All this assumes that the purchaser has correctly apportioned their management and capital costs to their community activities and is able to hire the extra staff. Under a fixed price HRG system, as in England, the Obstetric unit then gets paid a lower price based on national-average costs. The Obstetric unit then becomes enriched in more complex cases and even more expensive to run, and will take steps to reduce costs, possibly increasing neonatal admissions. The actual system-wide cost will have increased, and patient satisfaction will have declined.

While an HRG/DRG tariff will always be required to pay for activity which occurs outside of administrative boundaries, it is important to recognize that any HRG/DRG tariff is a gross distortion of the real costs. While the DRG/HRG tariff cost is real to the purchaser, it does not in any way reflect the real behaviour of costs in individual maternity units. For example, most maternity units operate at a fixed staffing level (and cost) irrespective of the number of births. It is only when births go above a certain threshold that staffing is increased. The existence of the cycle for births arising out of the WWII baby boom in the UK presents real challenges to the required bed and midwife numbers, and hence, real costs. As it has been demonstrated in the spreadsheet in Appendix A, such changes in births disproportionately affect the misalignment between real costs and tariff-based income based on where the maternity unit is located. It is strongly suggested that both purchasers and maternity units obtain a finance department input regarding the real-world behaviour of costs before making decisions based on tariff prices. Tariff prices are subject to high year-to-year volatility and can change with refinements to the tariff structure. See O13, O17–19 in [1].

A systematic review of maternity costs concluded that there was insufficient evidence to show that Midwife-led care had lower costs [38].

Another study observed that home births can only be cost-effective if the midwives are organized into larger groups, or if they work for hospitals that also facilitate home births. A model in which midwives work separately or in pairs to assist with a home birth and are on call for one birth at a time is not cost-effective [9]. Appropriate overhead apportionment may make home births even more expensive than they appear.

A systematic review of the cost-effectiveness of alternative models of maternity care showed weak evidence that any were cost saving relative to traditional care. However, this made no reference to the effect of size on occupancy, turn-away and costs. This study highlighted the need for more research incorporating appropriate models and population diversity [128]. An Australian study demonstrated nearly 50% variation in the cost of the first 1000 days after birth between different maternity systems in Queensland [129]. Hence, any process of system changes such as that proposed in the NHS England ‘Maternity transformation program’ must be subject to rigorous local evaluation and system-wide costing before unanticipated consequences are institutionalized, and the resulting inertia locks such consequences into the system.

### 4.18. Key Recommendations

The following key recommendations arise from this study.
Government health departments should encourage the use of turn-away for understanding maternity unity capacity preparedness.There is reliable evidence that maternity demand is subject to hourly, seasonal and environmental fluctuations, implying that the annual average occupancy should ideally be below 0.01% turn-away.Any maternity unit with an annual average turn-away greater than 1% must flag this on the hospital risk register and implement plans to correct this situation.Research is required to disentangle the effects of turn-away and poor staffing on safety and outcomes in maternity units.Maternity units should monitor bed occupancy and associated turn-away hourly throughout the year in the birthing unit, the postpartum maternity unit, any associated maternity (short stay) assessment unit and any Midwife-led community units. Past trends in such metrics should also be investigated to determine the local fluctuations in demand and ongoing trends. A chart showing daily admissions or births over many years is always a helpful tool.Maternity units should refresh their estimates of future demand every two to three years and compare how actual demand compares with past estimates.Government regulators should establish guidelines regarding the maximum acceptable turn-away in maternity units.Benchmarking of maternity unit minimum acceptable LOS needs to be against nationally agreed levels of quality and safety. High turn-away units should be excluded from the derivation of such benchmarking.To compare costs on a like-for-like basis, the cost per HRG/DRG requires the identification of the separate components of cost, namely, depreciation on capital (buildings and equipment), organization-wide apportioned costs for all the non-patient facing departments and the direct costs of care. The direct costs of care per birth will be higher as the unit gets smaller and, on this basis, small midwife-led low risk units are unlikely to be cost effective—although they may be considered desirable by mothers.In England, which uses the HRG tariff, all maternity units should receive extra funding based on size to mitigate the unavoidable higher costs relating to smaller size.Research is required as to how maternity units cope on days when there are unusually high births arising from Poisson variation, i.e., do they resort to emergency C-section to cope? Due to the skew in the Poisson distribution, this becomes a bigger problem as the unit gets smaller.As an extension of #11, the safe staffing levels for maternity units must be formulated with a higher ratio as size decreases.

### 4.19. Limitations of the Study

The major limitation of this study is that no data on bed numbers and average occupancy are available for midwife-led units, and that real time LOS is not reported for the English NHS; although, a reasonable approximation was made in this study. The study attempts to direct maternity managers toward a structed approach to create scenarios for future demand. There is never a ‘right’ answer, only a series of potential outcomes. Examples of trends in England are merely illustrative of the principles. Maternity managers may need to obtain help from local authorities regarding past and future trends in new house construction, numbers of asylum seekers, etc. Local or regional Public Health teams may be able to supply births at electoral wards or other relevant geographic areas. The optimum LOS is likewise subject to a degree of subjectivity and will depend greatly on the local availability of community midwives—always be conservative.

## 5. Conclusions

This study has established that forecasting maternity demand is fraught with uncertainty along with high levels of volatility, due to size and the local environment. Given that maternity services must operate at low turn-away, the high uncertainty and volatility dictates that capacity planning must be conducted for the worst-case scenario, which will be unique to each location. Migration and local house building play a huge role in births and the associated uncertainty. Migrants tend to congregate in large cities, which amplifies the location specificity of future demand. Judging by the situation in England there are no standards regarding an acceptable maximum level of turn-away, and indeed widespread ignorance regarding this issue appears to prevail. It is highly likely that high turn-away maternity units have artificially low LOS and have hidden costs for premature discharge and wider patient harm. High uncertainty in the government statistical agency forecasts of births have been present for many years and do not appear to have triggered any scrutiny of the methods and their hidden assumptions. The national strategy regarding midwife training seems based on simple assumptions. Each country needs to define an optimum postpartum average LOS for each type of birth which avoids unnecessary neonatal admissions. For the entirety of Obstetric unit care this optimum (including time in the birthing unit and admissions for conditions during pregnancy) is probably around 2.1 to 2.5 days (real time LOS) and for midwife units around 1.5 to 1.6 days (depending on the proportions of admissions during pregnancy but not birth which occur in the Obstetric versus Midwife units). All dependent on the age and parity of the mother, obesity, social group, etc. Each country will echo aspects of the situation seen in England.

On a like-for-like basis, small midwife-led community units are likely not financially viable. However, this does not mean they should not be available. Such units can operate above a 1% turn-away since those about to give birth can be diverted to the nearest larger Obstetric unit should the midwife unit be at full capacity. This requires modelling to determine the exact figure for optimum turn-away, which will depend on the respective sizes of the midwife and Obstetric units. From a statistical perspective, maternity units are small and subject to highly nonlinear adverse effects of size (Figure 2). Size dictates entirely unavoidable capital and staffing costs, and in England the HRG tariff must include adjustment to ensure equity and fairness. The responsibility for such equity and fairness resides entirely with the DHSC and NHS England, as indeed with all government healthcare bodies around the world.

## Figures and Tables

**Figure 1 ijerph-22-00087-f001:**
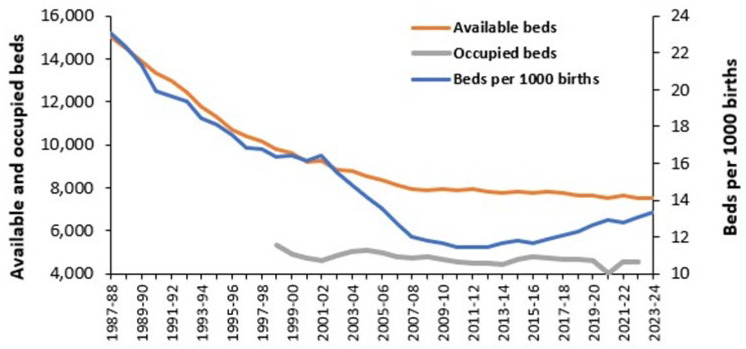
Trend in available maternity beds in Consultant-led Obstetric units and the ratio of available beds per birth in England, 1987/88 to 2023/24. Data from [11,14,17,21].

**Figure 2 ijerph-22-00087-f002:**
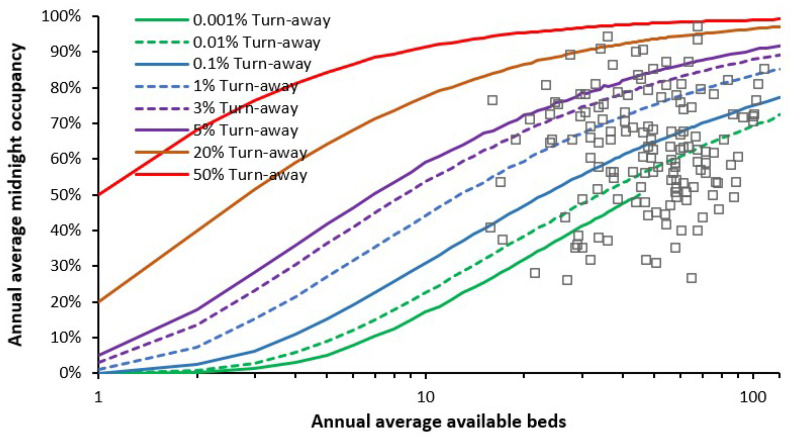
English NHS maternity unit average available beds, average occupancy and calculated turn-away during 2023/24 [11].

**Figure 3 ijerph-22-00087-f003:**
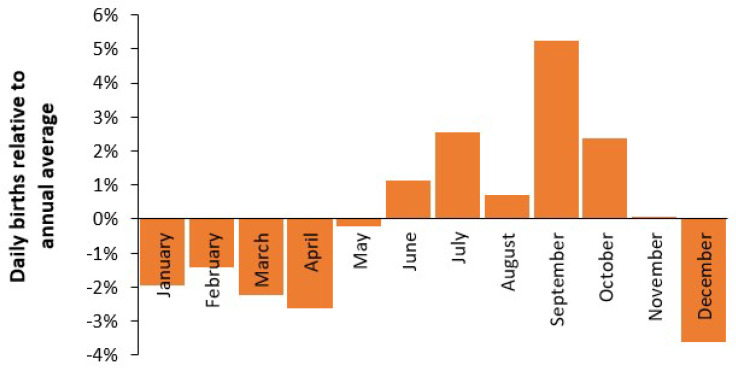
Average daily live births in England and Wales (2010 to 2022) relative to the annual average [14]. Live births can be skewed by discretionary C-sections.

**Figure 4 ijerph-22-00087-f004:**
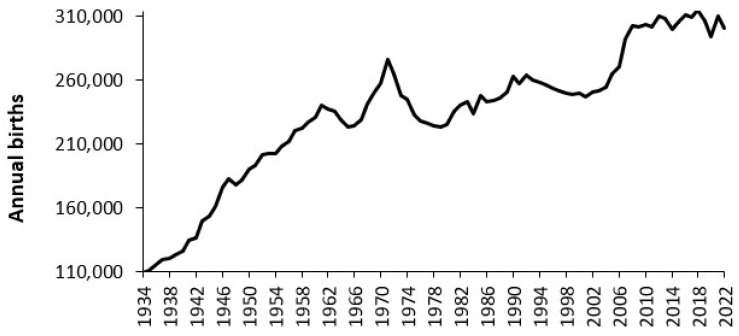
Trend in annual live births in Australia, 1934 to 2022. Data from [13].

**Figure 5 ijerph-22-00087-f005:**
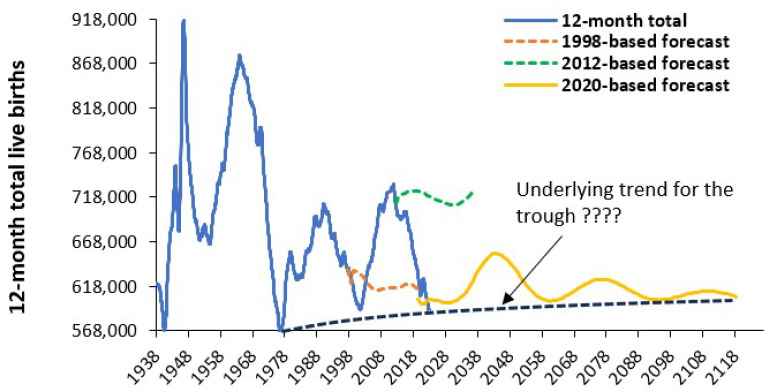
Trend in a moving 12-month total of live births in England and Wales, 1938 to 2023 [14], with three forecasts for future births made by the ONS [18].

**Figure 6 ijerph-22-00087-f006:**
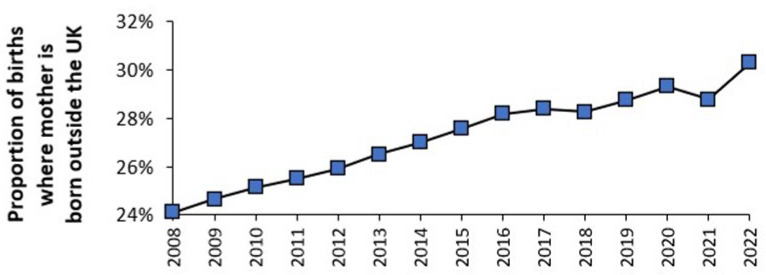
Trend in the proportion of annual births (England and Wales) in which the mother was born outside the UK. Data are from [15,16].

**Figure 7 ijerph-22-00087-f007:**
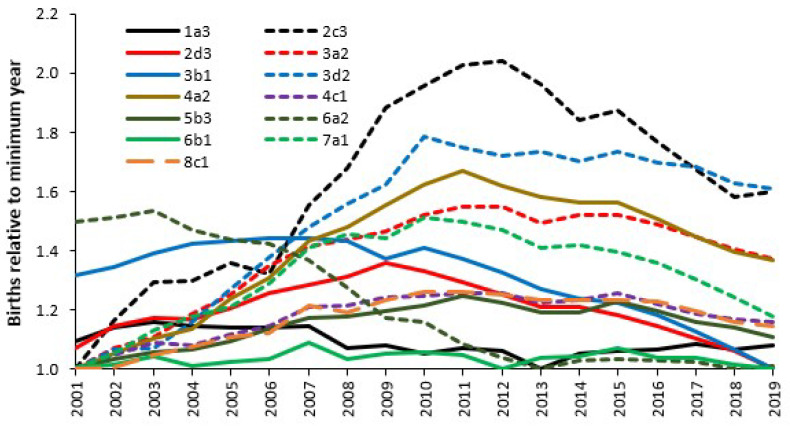
Trend in annual births (2001 to 2019) relative to the minimum year (mostly 2019) in England, using several illustrative social groups. Data from [19].

**Figure 8 ijerph-22-00087-f008:**
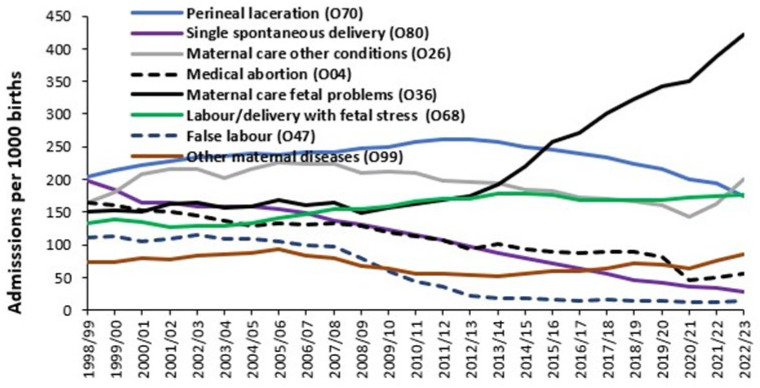
Trend in admissions per 1000 births for eight high volume maternity diagnoses in England. Data from [14,21].

**Figure 9 ijerph-22-00087-f009:**
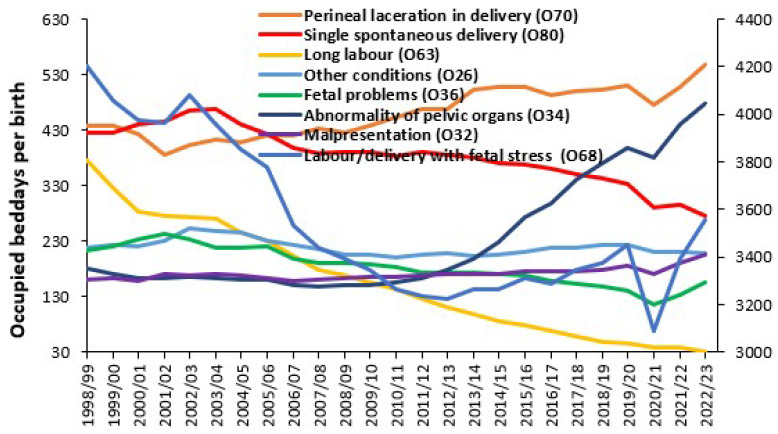
Trend in occupied bed days per 1000 births for eight high volume primary ICD-10 diagnoses in England. Data from [14,21]. O68 is right hand axis.

**Figure 10 ijerph-22-00087-f010:**
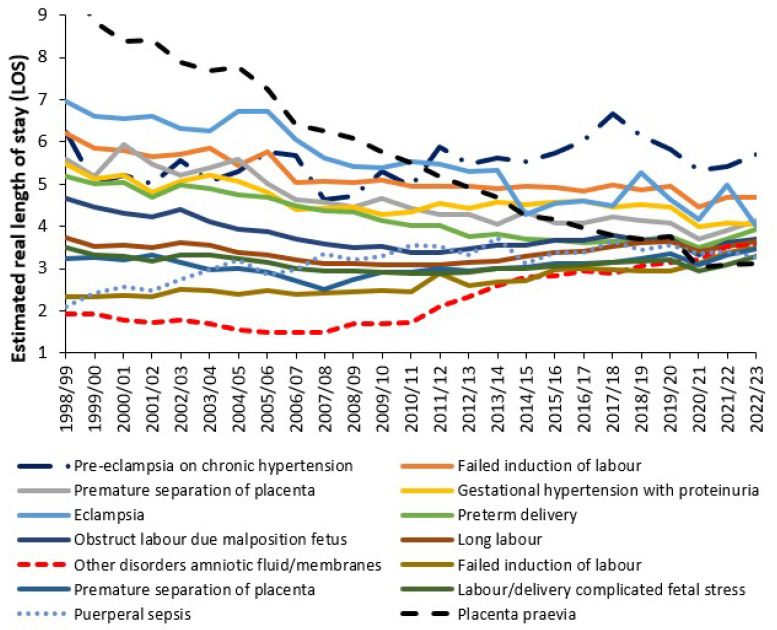
Trend in the estimated real average LOS for several ICD-10 primary diagnoses with the highest LOS among Chapter O diagnoses in 2022/23. Data from [21].

**Figure 11 ijerph-22-00087-f011:**
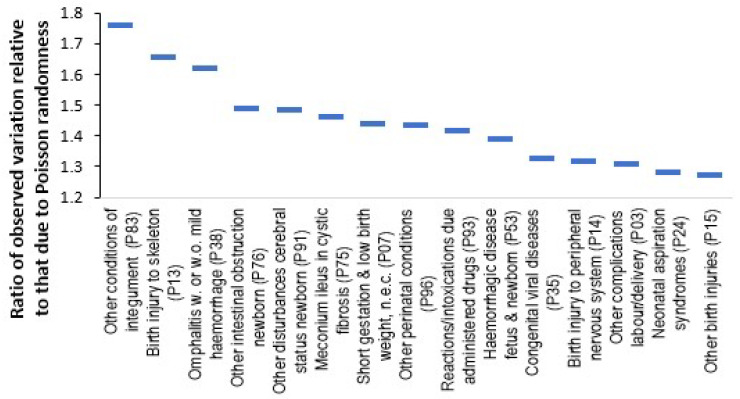
Top 15 neonatal conditions which are potentially affected by the environment and showing high variation in the year-to-year proportion females in each ICD-10 diagnosis in Chapter P. Data from [21]. n.e.c. = not elsewhere classified.

**Figure 12 ijerph-22-00087-f012:**
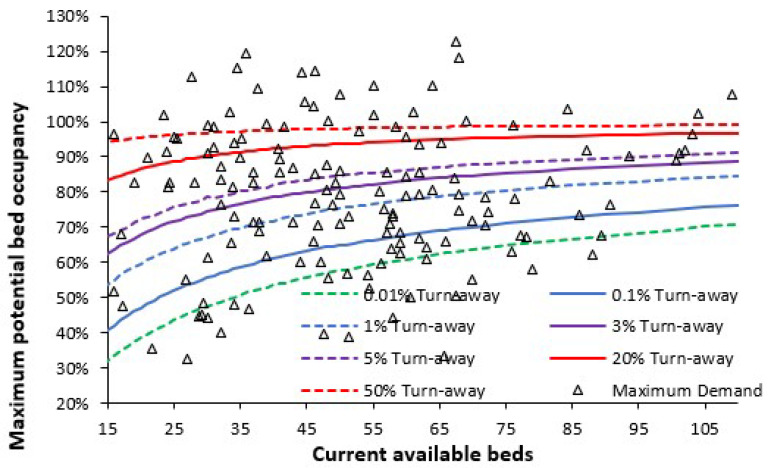
A stress test applied to the data in Figure 2 to illustrate the maximum possible effect of a high growth scenario upon bed occupancy with current bed numbers in England.

## Data Availability

All data are from publicly available sources. The data supporting various Figures is available on request.

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
