# Peer review of "Capacity Planning (Capital, Staff and Costs) of Inpatient Maternity Services: Pitfalls for the Unwary"

_ijerph, 2025, doi:10.3390/ijerph22010087_

Round 1

Reviewer 1 Report

Comments and Suggestions for Authors

I want to congratulate author on the manuscript and the work. It was a pleasure to read the manuscript. However, I suggest a more scientific writing tone so to guide the reader. The higher fixed costs of smaller units with higher chance of rate of return if occupancy being kept high, is a very interesting point to make. But it is buried. Consider direct hypothesis write up with evidence, as the structure. This is not a personal letter, thus, keep the recommendations based on data-driven work or prior works and surveys.

Discussion section is very long with repeated points. Please consider analyzing the results in fewer concise points.  

Minor suggestions: please consider revisiting the first paragraph and stick to proper referencing to prior body of the work with meaningful connection to the specific topic of this paper. 

I am personally very familiar with queuing theory, however, expanding on the concept can be helpful for the readers of this journal (i.e. how beds and midwives are servers in the theory). Similar comment goes to explaining the Erlang B concept, in addition to the history of equation. 

line 66: I think the paragraph would be very hard to follow for non-technical audience. consider re-writing, focusing on health language and providing modeling assumption and background as hypothetical explanation until the evidence is provided that indeed we are dealing with such arrival rates and distributions and so on. 

113-122: edit the format of formulations

Any references for 3.5%? Los of outpatient - any references? 

181 - still quiet a lot of assumptions that need data specific calibrations (efficiency of units, health workforce, complication variation). I would refrain from using the term "truly robust". Model robustness has a very specific meaning in optimization. 

188- move this motivation to intro and abstract. 

For birth rates: I suggest adding literature with regards to population forecasts. Although that is more complex, given the expected life expectency changes, but it might help with analyzing some of the observations. 

453 - how obesity is brought in, please keep the generic language such as complicated pregnancy risk factors such as x, y, z. 

Comments on the Quality of English Language

Please see my prior comments. I suggest a scientific write-up approach rather than open letter approach as it makes it hard to follow the opinions and personal observations from the historical trends. 

Author Response

Notes to the reviewers:

Many thanks for your time and thoughtful input into the study. Some further changes have been made in addition to your comments.

  1. A spreadsheet has been added as Supplementary material S2 detailing a simple method to estimate a likely maximum case scenario for births over the next 20 years. Four worked examples are given. The method can be adapted for countries other than England. Any manager can simply add both local and country/state/province/regional data to the spreadsheet.
  2. The method for calculating the real-world variation in the gender ratio for both neonatal and congenital admissions has been changed to reflect the fact that the number of admissions can change over time. Hence, the median admissions have been replaced by the average admissions for each of the paired years. The difference in the gender ratio is then converted to a standard deviation equivalent for each of the paired years (square root of the average of the two years). The median of the real-world standard deviations is then calculated for each diagnosis. This is a better method which allows all available years to be used and alters the order for the diagnoses (See revised Figure 11). The median overcomes any problems with changes in coding which may occur over time or as a sudden change. This has been added to the Methods section.
  3. A section has been added discussing whether governments should stipulate a minimum acceptable length of stay for common situations such as C-section, etc. A detailed analysis of trends in LOS by ICD-10 diagnosis is now given in Supplementary material S1. 
  4. A new results section has been added regarding possible circadian patterns in admissions and LOS. This section is based on anonymized real-time data kindly provided by a large maternity unit. Patterns in both 24-hour cycle in admissions and LOS can be discerned for patients staying less than 24 hours or >24 hours. It is unclear how staffing patterns may affect these results. However, it emphasizes the futility of capacity planning based on bland averages.
  5. A short paragraph has been added regarding how an HRG/DRG tariff can create financial imbalance since costs in the real-world of a maternity unit behave very differently.
  6. A section has been added explaining why the ONS process for estimating future births looks to contain a hidden flaw and how this could be remedied.
  7. Have added another 10 relevant references.

Reviewer 1 (specific actions)

I want to congratulate author on the manuscript and the work. It was a pleasure to read the manuscript. However, I suggest a more scientific writing tone so to guide the reader. The higher fixed costs of smaller units with higher chance of rate of return if occupancy being kept high, is a very interesting point to make. But it is buried. Consider direct hypothesis write up with evidence, as the structure. This is not a personal letter, thus, keep the recommendations based on data-driven work or prior works and surveys.

Discussion section is very long with repeated points. Please consider analyzing the results in fewer concise points.

I have edited the discussion and removed as many repeated points as possible. I have left the results as they are since the idea is to fully convey the degree of uncertainty in maternity capacity planning. My audience is largely hospital managers, midwives and policy makers, who may benefit from understanding the profoundly important role of uncertainty in capacity planning.

Minor suggestions: please consider revisiting the first paragraph and stick to proper referencing to prior body of the work with meaningful connection to the specific topic of this paper.

When I wrote the first paper I agreed with the Editor-in-Chief to take this approach to avoid self-citation.

I am personally very familiar with queuing theory, however, expanding on the concept can be helpful for the readers of this journal (i.e. how beds and midwives are servers in the theory). Similar comment goes to explaining the Erlang B concept, in addition to the history of equation.

The Introduction has been expanded accordingly

line 66: I think the paragraph would be very hard to follow for non-technical audience. consider re-writing, focusing on health language and providing modeling assumption and background as hypothetical explanation until the evidence is provided that indeed we are dealing with such arrival rates and distributions and so on.

113-122: edit the format of formulations

IJERPH will edit to their style during the final proofing stage

Any references for 3.5%? Los of outpatient - any references?

A section has been added to explain. This was roughly based on the analysis of real-time data from a single Obstetric unit and the chosen formula is a compromise to approximate a national average covering both Obstetric and Midwife units. As mentioned in the study all units need to calculate LOS figures based on real-time data.

181 - still quiet a lot of assumptions that need data specific calibrations (efficiency of units, health workforce, complication variation). I would refrain from using the term "truly robust". Model robustness has a very specific meaning in optimization.

The words ‘truly robust’ have been replaced by the word ‘common’. The factors you have mentioned do not affect the turn-away per se but apply to the level of occupied beds relative to the available beds. However, comments have been added to the Length of Stay section regarding the necessity for a learning culture to minimize adverse events and thus reduce the occupied bed days via best practice.

188- move this motivation to intro and abstract.

A comment has been added into the abstract.

For birth rates: I suggest adding literature with regards to population forecasts. Although that is more complex, given the expected life expectency changes, but it might help with analyzing some of the observations.

A section has been added. The Office for National Statistics insist that population forecasts are treated as estimates and NOT as predictions. The English NHS has largely fallen into the trap of treating them as predictions!

453 - how obesity is brought in, please keep the generic language such as complicated pregnancy risk factors such as x, y, z.

Appropriate wording has been added.

Comments on the Quality of English Language

Please see my prior comments. I suggest a scientific write-up approach rather than open letter approach as it makes it hard to follow the opinions and personal observations from the historical trends.

I have endeavoured to do this, however, a degree of comment based on 30-years of experience is sometimes required to make sense of why and how decisions are made. Hence, on occasions, the formulation of personal suggestions.

Should I have missed anything please let me know.

Once again may thanks for your valuable input which is greatly appreciated.

Reviewer 2 Report

Comments and Suggestions for Authors

Dear author

Thank you for the opportunity to read the manuscript, which I read with great interest.

The manuscript has an interesting theme, however, it needs some changes that will significantly improve it. Below you will find some points in the manuscript that need clarification, refinement, re-analysis, re-writing and/or additional information and suggestions on what can be done to improve it.

Title - Adequate

Abstract - I suggest organizing the abstract by background, objective, method, results and conclusions. Some descriptors should be revised and brought into line with DeCS/Mesh: birth - correct for Parturition; length of stay - correct for Length of Stay; staffing - correct for Workforce; quality of care - correct for Quality of Health Care; healthcare policy - correct for Health Policy. The remaining Keywords are not DeCS/Mesh descriptors, which may not make it easy to search the databases for citations of the article.

Section 1 (Introduction) - this section needs some adjustments, as some information and/or points are missing or unclear, and should be included or better written, I will present some items:

o Why is this research important?

o What problem does this research address?

o What is the aim of the study?

o What are the research questions?

Section 2 (Materials and Methods) - in this section some points should be clarified and improved and included, namely:

o The Methods section requires reorganization with clear subheadings to improve readability, namely, study design sample type....

o Clarification of the research criteria...

o What statistical protocol was used, margin of error...

Section 3 (Results) - This section leaves me with doubts, because without some of the missing methodological information mentioned above, I can't clearly evaluate the data.

o Clarify some concepts, namely “Consultant-led Obstetric units” and “midwife-led units” since this is not the reality in all countries.

Section 4 (Discussion) - the discussion leaves me with doubts, as I don't know the specific methodological issues and I can't make a correct assessment, but it seems reasonable to me.

Section 5 (conclusion) - Not knowing the objective and the research questions, I can't make a proper evaluation.

The references seem adequate to me, however there are many references to supplementary material and to the reference: Jones, R. Addressing the knowledge deficit in hospital bed planning and defining an optimal region for the number of 1121 different types of hospital beds in an effective health care system. Int. J. Environ. Res. Public Health 2023, 20, 7171. 1122 https://doi.org/10.3390/ijerph20247171 which may cause some confusion.

Author Response

Notes to the reviewers:

Many thanks for your time and thoughtful input into the study. Some further changes have been made in addition to your comments.

  1. A spreadsheet has been added as Supplementary material S2 detailing a simple method to estimate a likely maximum case scenario for births over the next 20 years. Four worked examples are given. The method can be adapted for countries other than England. Any manager can simply add both local and country/state/province/regional data to the spreadsheet.
  2. The method for calculating the real-world variation in the gender ratio for both neonatal and congenital admissions has been changed to reflect the fact that the number of admissions can change over time. Hence, the median admissions have been replaced by the average admissions for each of the paired years. The difference in the gender ratio is then converted to a standard deviation equivalent for each of the paired years (square root of the average of the two years). The median of the real-world standard deviations is then calculated for each diagnosis. This is a better method which allows all available years to be used and alters the order for the diagnoses (See revised Figure 11). The median overcomes any problems with changes in coding which may occur over time or as a sudden change. This has been added to the Methods section.
  3. A section has been added discussing whether governments should stipulate a minimum acceptable length of stay for common situations such as C-section, etc. A detailed analysis of trends in LOS by ICD-10 diagnosis is now given in Supplementary material S1.
  4. A new results section has been added regarding possible circadian patterns in admissions and LOS. This section is based on anonymized real-time data kindly provided by a large maternity unit. Patterns in both 24-hour cycle in admissions and LOS can be discerned for patients staying less than 24 hours or >24 hours. It is unclear how staffing patterns may affect these results. However, it emphasizes the futility of capacity planning based on bland averages.
  5. A short paragraph has been added regarding how an HRG/DRG tariff can create financial imbalance since costs in the real-world of a maternity unit behave very differently.
  6. A section has been added explaining why the ONS process for estimating future births looks to contain a hidden flaw and how this could be remedied.
  7. Have added another 10 relevant references.

Reviewer 2 (specific actions)

Thank you for the opportunity to read the manuscript, which I read with great interest.

The manuscript has an interesting theme, however, it needs some changes that will significantly improve it. Below you will find some points in the manuscript that need clarification, refinement, re-analysis, re-writing and/or additional information and suggestions on what can be done to improve it.

Title - Adequate

Abstract - I suggest organizing the abstract by background, objective, method, results and conclusions.

MDPI allow both approaches to the Abstract. Given the audience of hospital managers,  midwives and policy makers I chose the text approach as it is easier to read.

Some descriptors should be revised and brought into line with DeCS/Mesh: birth - correct for Parturition; length of stay - correct for Length of Stay; staffing - correct for Workforce; quality of care - correct for Quality of Health Care; healthcare policy - correct for Health Policy. The remaining Keywords are not DeCS/Mesh descriptors, which may not make it easy to search the databases for citations of the article.

Thank you for this valuable suggestion it is greatly appreciated. Changes have been made. The remaining words are those which a hospital manager may use in a general internet search.

Section 1 (Introduction) - this section needs some adjustments, as some information and/or points are missing or unclear, and should be included or better written, I will present some items:

As some background, during writing Part 2 of the series I collected maternity bed numbers and occupancy data along with the same for paediatrics and realized that the situation was very poor. I had written a piece back in 2012 warning about the cycle in births arising from the WWII baby boom of which NHS England was clearly ignorant or just ignored. I realized that history was likely to repeat itself, so the research grew out of this…… alas without a set study design other than the aim to avoid another maternity capacity fiasco likely to occur at some time around 2035. If you like the research design was to give enough information for maternity units to make sensible capacity planning decisions and to explain why the worst-case scenario is a valid benchmark. The pressing need was to act quickly since the planning cycles for midwife training and capital investment takes such a long time. As it were warts and all.

o Why is this research important?

o What problem does this research address?

o What is the aim of the study?

o What are the research questions?

The study is written in a particular style to be readable by managers and midwives. These aims are now shown in the Introduction.

Section 2 (Materials and Methods) - in this section some points should be clarified and improved and included, namely:

o The Methods section requires reorganization with clear subheadings to improve readability, namely, study design sample type....

o Clarification of the research criteria...

o What statistical protocol was used, margin of error...

The methods section has been changed. The samples are named and are usually births/admissions in England or England and Wales – depending on data availability.  Number of admissions per year for Chapter’s O (>1 million), P (>150,000) and Q (>66,000) are given There are two additional samples kindly provided by two English maternity units which are used to illustrate several points. Sample size is given on both occasions and is suitably large with 9,100 consecutive admissions in the smaller data set.

Section 3 (Results) - This section leaves me with doubts, because without some of the missing methodological information mentioned above, I can't clearly evaluate the data.

Sample sizes and relevant information has now been included.

A section added regarding how to use annual and quarterly data regarding bed occupancy and turn-away. As would be expected turn-away does flex during the year.

o Clarify some concepts, namely “Consultant-led Obstetric units” and “midwife-led units” since this is not the reality in all countries.

This has been clarified.

Section 4 (Discussion) - the discussion leaves me with doubts, as I don't know the specific methodological issues and I can't make a correct assessment, but it seems reasonable to me.

Now corrected as above.

Section 5 (conclusion) - Not knowing the objective and the research questions, I can't make a proper evaluation.

Hopefully clarified.

The references seem adequate to me, however there are many references to supplementary material and to the reference: Jones, R. Addressing the knowledge deficit in hospital bed planning and defining an optimal region for the number of 1121 different types of hospital beds in an effective health care system. Int. J. Environ. Res. Public Health 2023, 20, 7171. 1122 https://doi.org/10.3390/ijerph20247171 which may cause some confusion.

An agreement between myself and the Editor-in-chief to limit self-citation across the series of three papers.

I have had 56 views of the preprint with no comments to that effect. I guess people can always email if they need to talk.

Should I have missed anything please let me know.

Once again may thanks for your valuable input.
